# Uncovering Grounding IDs: How External Cues Shape Multimodal Binding

## Abstract

Large vision–language models (LVLMs) show strong performance across multimodal benchmarks but remain limited in structured reasoning and precise grounding. Recent work has demonstrated that adding simple visual structures, such as partitions and annotations, improves accuracy, yet the internal mechanisms underlying these gains remain unclear. We investigate this phenomenon and propose the concept of *Grounding IDs*, latent identifiers induced by external cues that bind objects to their designated partitions across modalities. Through representation analysis, we find that these identifiers emerge as consistent within-partition alignment in embedding space and reduce the modality gap between image and text. Causal interventions further confirm that these identifiers mediate binding between objects and symbolic cues. We show that Grounding IDs strengthen attention between related components, which in turn improves cross-modal grounding and reduces hallucinations. Taken together, our results identify Grounding IDs as a key symbolic mechanism that explains how external cues enhance multimodal binding and offer both interpretability and practical improvements.

## 1 Introduction

Large vision–language models (LVLMs), such as LLaVA (Liu et al., 2023b), GPT-4V (Achiam et al., 2023), and Qwen-VL (Bai et al., 2023) have achieved strong results in multimodal tasks, including image captioning, visual question answering, and embodied tasks. Despite this progress, they remain limited in structured reasoning (Campbell et al., 2024), precise grounding, and hallucination (Liu et al., 2024). Very recent studies suggest that external visual structure can systematically improve LVLM reasoning. For instance, Rudman et al. (2025) reveal that many LVLMs struggle with basic polygon identification, while adding shape annotations substantially boosts accuracy. Notably, *VISER* (Izadi et al., 2025) introduces more general visual structure (e.g., grids or horizontal lines) and demonstrates consistent gains across diverse reasoning tasks. Despite compelling advancements, the interpretability and mechanism-level understanding of this phenomenon remain underexplored. Prior work has largely focused on **what** visual structure can enhance reasoning, but not **how** it helps.

To bridge this critical understanding gap, we introduce **Grounding IDs**, symbolic multimodal identifiers induced by external cues. Our central hypothesis is that providing LVLMs with external visual cues (e.g., row labels, symbols, grid lines) aligned with textual cues (e.g., prompts for symbol-based scanning) leads LVLMs to generate latent identifiers that bind objects to their corresponding cues. These Grounding IDs propagate through embeddings and attention, enabling consistent correspondence within and across modalities. We validate this hypothesis through several analyses. First, observational probes show that Grounding IDs shape attention, reinforcing partition-level dynamics across modalities. Alignment experiments further reveal that symbolic cues strengthen within-partition binding and reduce the modality gap. Second, causal interventions confirm that external cues become causally bound to multimodal object tokens through Grounding IDs.

Beyond interpretability, we extend prior work on simple scaffolds (e.g., horizontal lines) to more effective multimodal cues, which align input modalities through a set of unique marks. Our ablation studies show that both visual and textual cues contribute to binding, with their combination yielding the greatest improvements in cross-modal grounding. Finally, we demonstrate the practical utility of Grounding IDs by showing that enhanced cross-modal attention substantially reduces hallucinations

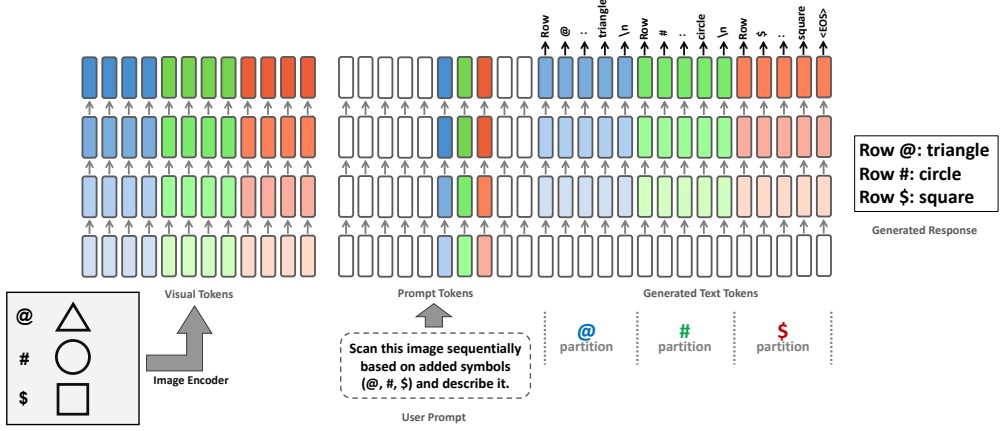

Figure 1: Conceptual overview of **Grounding IDs**. An input image is augmented with simple visual cues (e.g., {@, #, $}) and paired with a prompt that explicitly includes these symbols. Embeddings with the same Grounding IDs are displayed in matching colors across modalities, illustrating the reinforced binding between partitions and their corresponding textual descriptions.

in LVLMs. Overall, our results identify Grounding IDs as a core mechanism of partition-based binding, providing both mechanistic insight into LVLM reasoning and also practical gains on visual reasoning tasks.

## 2 BACKGROUND AND HYPOTHESES

Earlier studies show that adding external artifacts, combined with chain-of-thought (CoT) style prompting, can substantially enhance the reasoning abilities of vision–language models by shifting them from one-pass perception toward more systematic, system-2-like processing. Rudman et al. (2025) demonstrated that LVLMs are often "shape-blind": their vision encoders cluster frequent shapes but fail to distinguish less common ones, leading to errors in side counting and geometric reasoning. Explicit cues, such as annotated edges, encourage more deliberate strategies and yield large accuracy gains.

*VISER* (Izadi et al., 2025) generalizes this idea beyond polygons by introducing versatile and input-agnostic visual structure, such as horizontal lines, paired with sequential scanning prompts that promote structured reasoning. This approach reduces feature binding errors, encourages serial scene parsing, and consistently improves performance across reasoning tasks like counting and visual search. Building on this foundation, our work probes the internal circuits through which simple external artifacts shape visual reasoning in LVLMs.

Recent advances in mechanistic interpretability show that LLMs solve entity–attribute binding using **Binding IDs**, latent vectors that link entities with their attributes (Feng & Steinhardt, 2023). Causal mediation analyses support this finding and further show that these identifiers behave additively in representation space. Follow-up work extends this idea to vision–language models, showing that they employ similar vectors linking visual objects with textual references (Saravanan et al., 2025), with properties such as factorizability and position independence. These works examine binding between items and attributes in the standard setting. Inspired by these abstract identifiers, we investigate the underlying mechanism by which external visual structures, equipped with aligned textual cues, enhance LVLMs' reasoning.

In this work, we aim to answer a key open question: *why do external cues improve reasoning in LVLMs?* To this end, we introduce a general setting where both images and prompts are augmented with simple shared cues (e.g., symbol characters) that partition the input into distinct regions. With these aligned multimodal cues, the models appear to generate abstract identifiers that bind objects to their respective partitions, supporting more systematic visual scanning. Fig. 1 illustrates how these identifiers emerge both in representations of partitioned areas and in the generated textual descriptions of those partitions. We refer to these identifiers as Grounding IDs, since this partition-based

binding enhances multimodal grounding. In particular, this study seeks to validate the following hypotheses:

- **Existence**: Augmenting the inputs with multimodal external cues induces Grounding IDs that propagate through embeddings and attention, establishing within-partition binding across modalities.
- **Modality Gap**: Grounding IDs reduce the alignment gap between image and text representations of corresponding tokens.
- **Causal Relation**: Grounding IDs link objects to external cues in a causal manner, amplifying embedding alignment and increasing attention between connected parts.

To evaluate these hypotheses, the remainder of the paper is organized as follows. In Section 3, we probe model representations and internal dynamics to provide empirical evidence for Grounding IDs, showing how they shape attention patterns and reduce the modality gap. Section 4 presents causal intervention experiments that directly test whether Grounding IDs mediate object–cue binding. Finally, Section 5 explores the practical implications, demonstrating that enhancing cross-modal binding through Grounding IDs reduces hallucinations in LVLMs.

# 3 EMPIRICAL EVIDENCE FOR EXISTENCE OF GROUNDING IDS

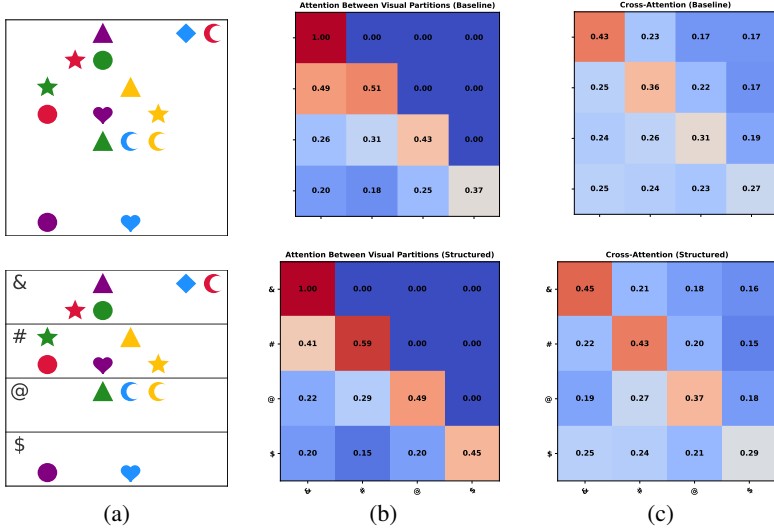

Figure 2: Illustration of attention patterns under baseline and structured inputs in the scene description task. (a) One dataset sample (top: baseline, bottom: structured). (b) Within-modality visual attention matrices. (c) Cross-modality attention matrices.

To investigate why external cues improve performance, we first analyze internal attention patterns of LVLMs. We focus on inference-time reasoning only, so we use a frozen 7B Qwen2.5-VL model without any fine-tuning. The tasks involve scene description and visual question answering, where the model produces detailed descriptions of the scene based on the provided input. The analysis is conducted on a synthetic dataset of 100 images. Each image contains 15 unique objects drawn from 35 shape–color combinations (7 shapes × 5 colors). To keep the analysis straightforward, each object occupies a single patch (28×28 pixels) and does not extend into adjacent patches. The baseline input follows the standard LVLM setup: the unmodified image, together with a plain scene-description prompt. The structured input augments this baseline by adding four symbols (&, #, $, @) to the image and prompt, and by dividing the image into four horizontal partitions with three lines, as shown in Fig. 2(a). Ablations of alternative cue designs are provided in Appendix A.4.

For the representation analysis, we first determine the textual tokens that correspond to visual objects after the model generates its output. Text tokens are matched to visual objects using regular-expression (regex) pattern matching, since each object type appears only once in the image. This

process assigns a partition label to each token in both modalities, allowing us to compute attention and embedding statistics in a partition-specific manner. Visual tokens are then grouped by their location within the four rows. Finally, we analyze attention patterns and embedding similarities within visual or cross-modal regions.

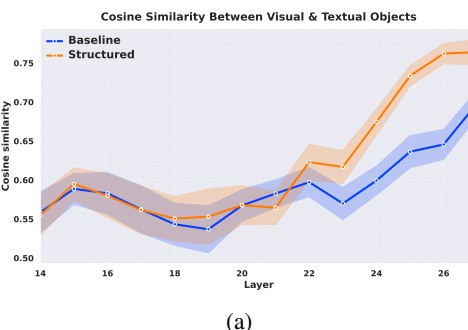
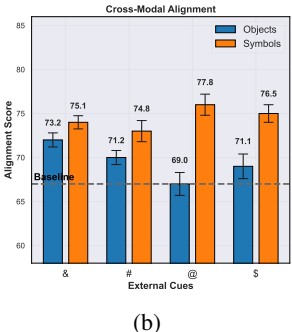

(a)                                                      (b)

Figure 3: Analysis of the modality gap. (a) Cross-modal alignment across layers, showing that improvements emerge in layers 22–27. (b) Average alignment in layers 22–27, reported separately for four partitions. Object embeddings under structured inputs achieve higher alignment than the baseline (dashed line), and symbol embeddings achieve even stronger alignment than objects.

**Attention Analysis.** For each token, we take the maximum attention score over all heads and then average across tokens within each partition (image rows), yielding a $4 \times 4$ matrix. Fig. 2 shows both within-modality and cross-modality attention matrices for baseline and structured inputs. Structured inputs exhibit stronger diagonal dominance, with attention concentrated within partitions. The same trend appears in cross-modal attention between visual and textual tokens, where structured inputs achieve stronger cross-attention than baselines. These results indicate that external cues induce systematic partition-based binding that enhances the model's attention.

**Modality Gap.** While attention captures binding structure, embedding similarity provides a complementary view of alignment between modalities. We measure cosine similarity between L2-normalized visual and textual embeddings corresponding to correctly generated object tokens. Similarities are computed layer-wise and averaged across samples. Both baseline and structured inputs show alignment strengthening in later layers (after layer 20). Structured inputs consistently achieve higher similarity, particularly in the last four layers, as shown in Fig. 3. This confirms that external cues reduce the modality gap in LVLMs by enhancing cross-modal alignment. In particular, cosine similarity between activation patches corresponding to external symbols across modalities is higher than that of dataset objects. Schrodi et al. (2024) report that post-hoc removal of gap-contributing dimensions, while effective in reducing the modality gap in VLMs, distorts the embedding geometry and degrades performance. In contrast, our results show that external multimodal cues reduce the modality gap through the model's internal dynamics without such disruptive interventions.

## 4 CAUSAL EVIDENCE OF GROUNDING IDs

The alignment and attention analyses in Section 3 provide correlational evidence that external cues induce partition-based alignment. Our hypothesis is that *Grounding IDs* act as abstract vectors that are induced to related tokens across modalities, increasing alignment and grounding. We now seek causal validation using a medium-sized LVLM, Qwen2.5-VL 7B. For causal mediation analysis, we use a simplified synthetic dataset in which each image contains four rows with one object per row. Rows are labeled with non-ordinal symbols $\{\&,\$,\#,@\}$ to prevent sequential order cues. The task is discriminative visual question answering: the prompt asks, for example, "What is in row @?", and the model must respond with the correct object description, such as "red circle."

To formalize notation, we define $\mathbf{o}_{s_i}^{s_j}$ as an object $\mathbf{o}$ that is *located* in the partition associated with symbol $s_i$ (subscript) and *bound* by the model to symbol $s_j$ (superscript). For example, $\mathbf{o}_{\&}^{\$}$ denotes an object positioned in the & partition but bound to $. For convenience, we use two shorthand cases:

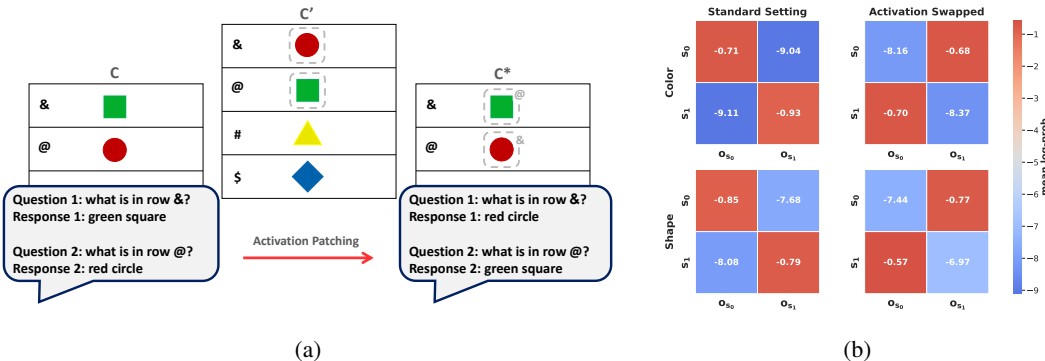

(a)                                                             (b)

Figure 4: Activation swap experiment. (a) Procedure in a case where source ($c'$) and target ($c$) contain the same objects. Activations from the & and @ partitions of $c'$ are patched into $c$, producing the patched context $c^*$. Predictions in $c^*$ follow the transferred bindings (gray) rather than host symbols. (b) Average log probabilities of $c$ and $c^*$ over valid row–symbol–object combinations. Rows and columns indicate the two selected query symbols and their corresponding objects.

- $\mathbf{o}_s^{\sim s}$ (adjacent object with $s$): an object located in the partition of $s$ but bound to a different symbol.
- $\mathbf{o}_{\sim s}^s$ (grounded object with $s$): an object bound to $s$ but located in a different partition.

## 4.1 ACTIVATION SWAPPING EXPERIMENT

We follow a causal mediation framework similar to prior studies (Vig et al., 2020; Feng & Steinhardt, 2023). We adopt standard terminology from mechanistic interpretability to describe the causal experiments in this section (Rai et al., 2024). We denote by $c$ an input *context* consisting of an image $x$ and a text prompt $p$. Given a model $M$ and layer $\ell$, let $h^{(\ell)}(c)$ be the hidden activations at that layer, including both visual patch tokens and textual tokens. In our interventions, we use three contexts: a *target* context $c$, a *source* context $c'$, and a *patched* context $c^*$ obtained by replacing a subset of activations in $h^{(\ell)}(c)$ with those from $h^{(\ell)}(c')$ and then running the remaining layers of $M$ on the modified activations.

Two contexts, $c$ and $c'$ are randomly sampled, and their activations are extracted across all layers. We then select two random symbols (e.g., \$ and #) and swap the patch activations of their corresponding object between the two contexts. For instance, the object in row \$ of $c'$ is inserted into row # of $c$, while the object in row # of $c'$ is patched into row \$. We pass the patched input $c^*$ to the model and record its predictions for the selected symbols. Fig. 4a shows a simple case where the swapped objects are the same between $c$ and $c'$. We intentionally choose the swap strategy to avoid ambiguity arising from the coexistence of $\mathbf{o}_{\sim s}^s$ and $\mathbf{o}_s^s$.

By analyzing the predictions of the patched context $c^*$, we discover that predictions consistently follow the binding symbols of the swapped objects rather than the symbols present locally in the host context. In other words, for a given partition $s$, the model outputs $\mathbf{o}_{\sim s}^s$ (grounded object) instead of $\mathbf{o}_s^{\sim s}$ (adjacent object). For example, if the object in row @ of $c'$ is a green square and it is inserted into row & of $c$, the model reports a green square for row @ even though that object is physically located beside &. This indicates that the model disregards local cues and follows the transferred symbol–object binding for its final prediction (Fig. 4a).

To quantify this effect across the dataset, we consider all valid swaps (across possible objects, symbols, and locations) and measure average swap accuracy. For each valid swap of object patches between $c$ and $c'$, we evaluate the model on $c^*$ using a fixed set of query symbols $s \in \mathcal{S}$. If the model's answer for symbol $s$ in $c^*$ matches the grounded object $\mathbf{o}_{\sim s}^s$ (i.e., it follows the transferred binding rather than the locally present object $\mathbf{o}_s^{\sim s}$), we count the trial as correct. Swap accuracy is the fraction of such queries for which the model outputs $\mathbf{o}_{\sim s}^s$.

If we follow the standard setting without intervention and treat $\mathbf{o}_s^{(\cdot)}$ as the correct label, standard accuracy drops sharply from $\mathbf{1.00}$ to $\mathbf{0.02}$ after swapping. In contrast, if we treat $\mathbf{o}_{\sim s}^s$ as the correct

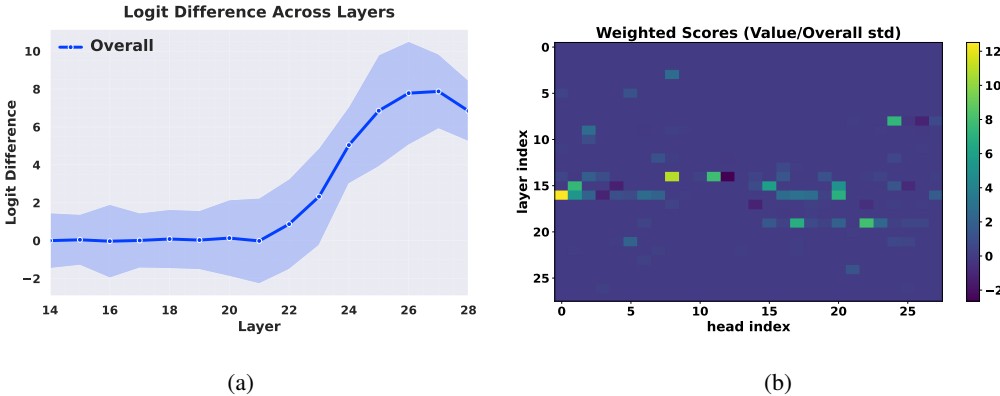

(a)                              (b)

Figure 5: Causal mediation analysis across layers. (a) Average logit differences between $\mathbf{o}_{\sim s}^{s}$ and $\mathbf{o}_{s}^{\sim s}$ across layers, showing where the model begins to favor the grounded object. (b) Signal-to-noise scores for attention differences between $\mathbf{o}_{\sim s}^{s}$ and $\mathbf{o}_{s}^{\sim s}$ patches across heads and layers.

label (i.e., prediction based on transferred bindings), swap accuracy remains high at **0.98**, confirming that predictions follow Grounding IDs; thus, object–symbol bindings are causally mediated by these IDs. Fig. 4b shows the average log-probability of predicted objects before and after activation swapping.

## 4.2 LAYERWISE ANALYSIS OF GROUNDING ID EMERGENCE

**Layerwise logit difference.** To locate where in the network the swapped grounding takes effect, we apply a logit-lens probe to the response token at each layer $\ell$. Let $L^{(\ell)}(x \mid c^*)$ be the unnormalized logit for token $x$ decoded from $h^{(\ell)}(c^*)$. We define the layerwise *logit difference*

$$\Delta L^{(\ell)} \;=\; L^{(\ell)}\big(\mathbf{o}_{\sim s}^{s} \mid c^*\big) \;-\; L^{(\ell)}\big(\mathbf{o}_{s}^{\sim s} \mid c^*\big),$$

averaged over all valid symbol–object pairs. Positive values of $\Delta L^{(\ell)}$ indicate that, at layer $\ell$, the representation favors the grounded object over the local object.

Next, we investigate in which layers the model begins to predict the intervened grounded object ($\mathbf{o}_{\sim s}^{s}$) rather than the adjacent object ($\mathbf{o}_{s}^{\sim s}$). The goal is to identify the layer at which the model's prediction shifts toward the object carrying the transferred binding in the patched context. In this experiment, we use a monochrome dataset for simplicity. We apply the logit lens technique (nostalgebraist, 2020) to the single-word response token at each layer $\ell$. Let $L^{(\ell)}(x \mid c^*)$ be the unnormalized logit for token $x$ decoded from $h^{(\ell)}(c^*)$. We define the layerwise logit difference

$$\Delta L^{(\ell)} \;=\; L^{(\ell)}\big(\mathbf{o}_{\sim s}^{s} \mid c^*\big) \;-\; L^{(\ell)}\big(\mathbf{o}_{s}^{\sim s} \mid c^*\big), \tag{1}$$

to compute the prediction tendency between $\mathbf{o}_{\sim s}^{s}$ and $\mathbf{o}_{s}^{\sim s}$ in each patched context, and then average across all valid symbol–object pairs.

As shown in Fig. 5a, this difference becomes positive in the later layers (20–27), indicating that the model increasingly favors the grounded object after the intervention. This result complements the representational trend in Fig. 3a, where alignment for structured inputs increases in the same higher layers, showing that the late-layer alignment shifts are accompanied by causal evidence of Grounding ID usage.

We also investigate which attention heads are most responsible for propagating Grounding IDs as opposed to relying on local visual proximity. For each head and layer, we compute the difference in attention from the response token to visual tokens corresponding to $\mathbf{o}_{\sim s}^{s}$ versus $\mathbf{o}_{s}^{\sim s}$ across samples. To quantify consistency, we divide the mean difference by its standard deviation, yielding a signal-to-noise ratio that reflects how reliably a head prefers grounded objects. To emphasize attention to meaningful regions, this ratio is multiplied by the head's average attention weight. Let $\alpha^{(\ell,h)}(\mathbf{r} \to \mathbf{o})$ denote the attention weight from the response token $\mathbf{r}$ to the patch of object $\mathbf{o}$ at layer $\ell$, head $h$. For

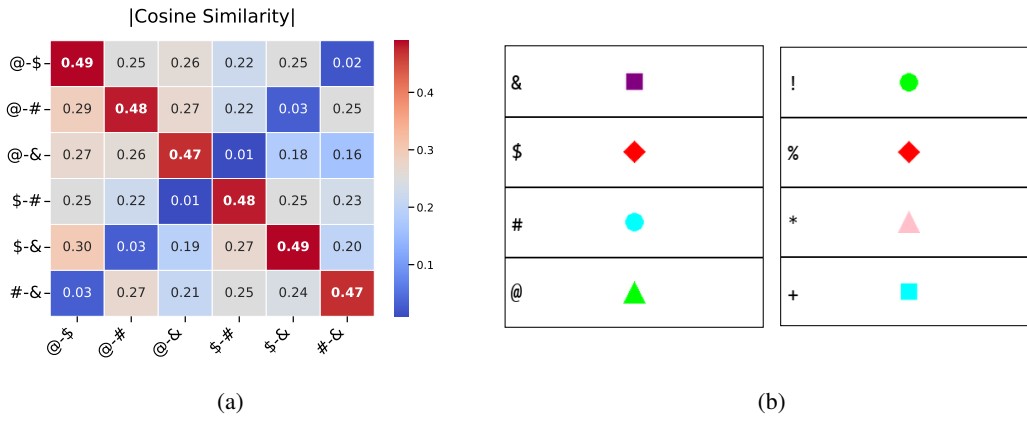

(a)           (b)

Figure 6: Characteristics of Grounding IDs. (a) Cosine similarity between averaged differential vectors of symbol patches (vertical) and their corresponding Grounding IDs (horizontal). (b) Disjoint-symbol control for the activation-swap test. The source context contains the symbols $\{\&,\$,\#,@\}$, while the target context contains a nonoverlapping set $\{!,\%,\times,+\}$.

each head, we compute

$$S^{(\ell,h)} \;=\; \frac{\mathbb{E}\big[\alpha^{(\ell,h)}(\mathbf{r}\to\mathbf{o}^s_{\sim s}) - \alpha^{(\ell,h)}(\mathbf{r}\to\mathbf{o}^{\sim s}_s)\big]}{\mathrm{Std}\big[\alpha^{(\ell,h)}(\mathbf{r}\to\mathbf{o}^s_{\sim s}) - \alpha^{(\ell,h)}(\mathbf{r}\to\mathbf{o}^{\sim s}_s)\big]} \cdot \mathbb{E}\big[\alpha^{(\ell,h)}(\mathbf{r}\to\text{image})\big], \qquad (2)$$

where expectations are taken over samples and valid symbol–object pairs. Heads with high $S^{(\ell,h)}$ consistently prefer grounded objects over merely co-located ones and are interpreted as key carriers of Grounding IDs. The resulting scores are visualized in Fig. 5b, showing that Grounding IDs cause significant attention shifts in certain heads, particularly in middle layers (layer 16), which attend more to grounded objects.

### 4.3 The Characteristics of Grounding IDs

Feng & Steinhardt (2023) shows that, in LLMs, Binding IDs linking entities and their attributes are largely context independent. In contrast, our causal mediation experiment demonstrates that Grounding IDs are directly predictable from their corresponding symbols. To probe this relationship, we analyze the relational structure of symbols and their induced Grounding IDs. Concretely, for each pair of symbols (e.g., &, #), we compute the difference between their symbol patch activations. For the same pairs, we also compute the difference between the corresponding object patch activations. Averaging these differential vectors across the dataset cancels out confounding factors such as shape and position, yielding two structured spaces: one defined by symbol differences and one by Grounding ID differences. Fig. 6a reports cosine similarities between these two spaces, revealing strong correspondence between the symbol space and the Grounding ID space.

To further assess this conjecture, we repeat the activation swap experiment but assign the target image a disjoint set of symbols (e.g., $+, \times, \%, !$) that do not overlap with those in the source (Fig. 6b). After activation patching, we query the model in $c^*$ using source symbols (e.g., &). Surprisingly, the model correctly outputs the object bound to the source symbol, even though no explicit occurrence of the symbol & is present in the host context. The average prediction accuracy reaches $0.86$, which is considerably higher than the random chance level. Overall, these experiments uncover the nature of Grounding IDs from complementary perspectives.

## 5 Behavioral Implications of Grounding IDs

Our observations in Section 3 exhibit a reduced modality gap and increased attention between related partitions in samples with external cues. The causal analysis in Section 4 further reveals the presence of latent identifiers that bind to the embeddings of the corresponding partitions. Together, these results describe a mechanism by which external cues improve visual reasoning: they slightly modify

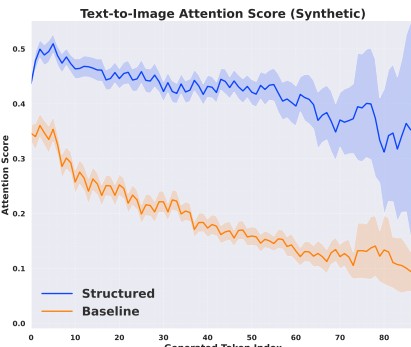

Figure 7: Averaged cross-attention behavior on the synthetic dataset with 20 objects. Attention is computed using a max operation with a window size of 5 and stride of 1 across generated tokens.

Table 1: Evaluation of the scene description task on synthetic datasets, each containing 500 samples with 10, 15, or 20 unique objects per image.

| # Obj. | Method | Precision | Recall | F1 | Accuracy |
|---|---|---|---|---|---|
| **10** | Baseline | 0.56 | 0.56 | 0.58 | 0.42 |
| | Structured (text-only) | 0.59 | **0.68** | 0.63 | 0.46 |
| | Structured (img-only) | 0.53 | 0.59 | 0.56 | 0.38 |
| | **Structured** | **0.74** | 0.58 | **0.65** | **0.48** |
| **15** | Baseline | 0.30 | 0.49 | 0.37 | 0.24 |
| | Structured (text-only) | 0.33 | **0.61** | 0.44 | 0.27 |
| | Structured (img-only) | 0.43 | 0.51 | 0.46 | 0.30 |
| | **Structured** | **0.67** | 0.53 | **0.59** | **0.46** |
| **20** | Baseline | 0.14 | 0.45 | 0.21 | 0.12 |
| | Structured (text-only) | 0.29 | 0.57 | 0.39 | 0.24 |
| | Structured (img-only) | 0.39 | 0.42 | 0.40 | 0.24 |
| | **Structured** | **0.65** | **0.59** | **0.62** | **0.40** |

the model's internal dynamics and equip visual embeddings with auxiliary identifiers that strengthen the model's grounding ability. To examine how this property affects downstream behavior, we first evaluate the impact of Grounding IDs on hallucination mitigation during long caption generation, since this task directly depends on grounding. We then assess the effect of aligned multimodal cues on broader visual reasoning tasks.

## 5.1 HALLUCINATION MITIGATION

In Section 3, we reported overall attention enhancement across the bound partitions. To test whether this partition-based effect also helps LVLMs remain focused on the image in long responses, we measure cross-attention during generation. Using the same synthetic dataset as in Section 3, the model is asked to produce detailed descriptions of each image based on the provided external cues. We compute cross-attention from generated tokens to image patches using a sliding window, taking the maximum value within each window of size 5. Plotting these values against token position (Fig. 7) reveals a consistent decline: tokens appearing later in the response attend less to the image, indicating that visual grounding diminishes as generation progresses. Importantly, the structured baseline shows both higher initial attention and a slower rate of decline compared to the baseline. This indicates that external cues sustain grounding over longer spans of text.

Prior work attributes hallucinations in LVLMs to the decay of visual attention and increasing reliance on language priors (Favero et al., 2024; Huang et al., 2024). Motivated by this, we evaluate hallucination on the controlled datasets from Section 3, measuring accuracy, precision, recall, and F1 for object descriptions, as reported in Table 1. Structured inputs consistently improve performance across all metrics, with particularly strong gains in precision, which is most relevant for avoiding mentions of nonexistent objects. For images with 10 objects, baseline recall is slightly higher, but as the number of objects increases, structured inputs outperform the baseline across all criteria. The performance gap due to external structure also widens as the number of objects increases. Notably, multimodal cues are considerably more effective than unimodal cues. These results highlight that sustained cross-modal grounding directly improves faithfulness of generated text.

Finally, we evaluate on large-scale hallucination benchmarks using MS-COCO images (Lin et al., 2014), with performance assessed by the hallucination-specific metric CHAIR (Rohrbach et al., 2018). We conduct experiments on two recent LVLMs, Qwen2.5-VL (Bai et al., 2025), and LLaVA-1.5 (Liu et al., 2023a). We observe that for natural scenes, simple horizontal lines are insufficient, and grid-based partitions provide more effective structure for sequential scanning of the image. To ensure visibility against diverse backgrounds, we add thin white margins around the scaffolding.

Results (Fig. 8 and Table 2) show substantial reductions in both CHAIR$_s$ and CHAIR$_i$ compared to baseline. Importantly, unlike most existing techniques for hallucination mitigation, the proposed approach is also applicable to black-box, closed-source models such as GPT-4o (Hurst et al., 2024) and

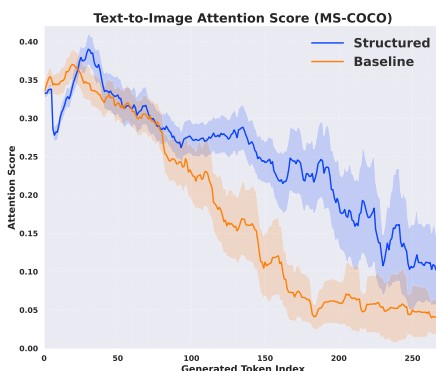

Text-to-Image Attention Score (MS-COCO)

Figure 8: Cross-attention behavior on the MS-COCO dataset across generated tokens, computed using a max operation with a window size of 10 and stride of 1.

Table 2: Evaluation on 500 MS-COCO samples using CHAIR metrics across open- and closed-source models. Results are reported for sentence-level (CHAIR$_s$) and instance-level (CHAIR$_i$) hallucination rates.

| Model | Method | CHAIR$_s\downarrow$ | CHAIR$_i\downarrow$ | Inference Time(s) |
|---|---|---|---|---|
| LLaVA-1.5 | Baseline | 51.60 | 13.20 | 3.41 |
| | Opera | 48.00 | 13.52 | 20.91 |
| | VCD | 54.40 | 14.28 | 7.81 |
| | SPARC | 55.20 | 12.78 | 4.50 |
| | **Structured** | **41.00** | **12.04** | 3.94 |
| Qwen2.5-VL | Baseline | 32.40 | 7.97 | 3.31 |
| | Opera | 29.60 | 10.76 | 23.50 |
| | VCD | 33.80 | 8.91 | 9.73 |
| | SPARC | 33.60 | 8.21 | 5.50 |
| | **Structured** | **27.20** | **5.36** | 6.04 |
| GPT-4o | Baseline | 29.20 | 6.40 | - |
| | **Structured** | **23.20** | **5.81** | - |
| Gemini2.5-Pro | Baseline | 44.20 | 8.64 | - |
| | **Structured** | **37.40** | **7.28** | - |

Table 3: Performance on counting and visual search benchmarks.

| | Counting Accuracy | | | Visual Search Accuracy | | |
|---|---|---|---|---|---|---|
| Model | Baseline | VISER | Grounding IDs | Baseline | VISER | Grounding IDs |
| **Qwen2.5-VL (3B)** | 30.00 | 37.83 | **43.00** | 0.00 | 37.83 | **45.96** |
| **Qwen2.5-VL (7B)** | 29.67 | 43.33 | **53.00** | 30.00 | 40.00 | **52.25** |
| **GPT-4o** | 10.50 | 26.50 | **32.33** | 49.41 | 73.40 | **80.62** |

Gemini-2.5-Pro (Comanici et al., 2025), which are considered strong models for visual reasoning. Notably, our simple strategy outperforms or matches specialized hallucination-mitigation methods such as VCD (Leng et al., 2024), OPERA (Huang et al., 2024), and SPARC (Jung et al., 2025), while requiring no additional inference modules and maintaining near-zero computational overhead. We also report results on another commonly used benchmark, POPE (Li et al., 2023), in Appendix A.6

## 5.2 VISUAL REASONING PERFORMANCE

We evaluate Grounding IDs on two visual reasoning benchmarks related to spatial grounding: counting and visual search. We conduct experiments on the datasets introduced in Izadi et al. (2025). The counting dataset consists of synthetic 2D scenes with 10 to 20 circles, increasing in steps of 2, with 100 images for each count level (600 total). The visual search dataset contains conjunctive search scenes with 20, 30, 40, and 50 objects. For each difficulty level, 100 images are created: 50 with the target present and 50 absent. Accuracy is computed separately for visible and invisible cases and combined using the harmonic mean.

We evaluate three models: Qwen-3B, Qwen-7B, and GPT-4o. Grounding IDs increase accuracy on both tasks and outperform the VISER baseline and the unstructured input. The gains depend on two conditions: each partition must have a distinct identifier, and the same identifiers must appear in the image and the prompt. The results are shown in Table 3.

## 6 RELATED WORK

**Interpretability.** Interpretability research investigates where and how information is represented and routed in neural networks. For **LLMs**, attention analyses study which tokens heads attend to and how this shapes predictions (Chefer et al., 2021; Clark et al., 2019), while circuit-oriented methods identify causal substructures and enable editing through activation or circuit discovery and factual localization (Conmy et al., 2023; Meng et al., 2022; Geva et al., 2023). Probing tools such as the *logit lens* decode intermediate states to reveal how token predictions evolve across layers (nostalgebraist, 2020). For **VLMs**, recent work explores where visual information is stored and how it is transferred into the language pathway (Basu et al., 2024), and adapts logit-lens–style spatial probes to study

grounding and localization inside multimodal models (Jiang et al., 2024; Neo et al., 2024). Overall, the interpretability methods provide the toolset we build on: causal interventions and layerwise probes to expose mechanisms underlying cross-modal binding and grounding.

**Abstract latent variables in representation space.** Recent advances suggest that models internally rely on abstract latent variables to maintain entity–attribute associations. In LLMs, *Binding IDs* are content-independent identifiers that link entities and attributes through a shared latent code; causal interventions that swap these vectors systematically change the inferred associations (Feng & Steinhardt, 2023). Follow-up work localizes a low-rank subspace that *causally* governs which entity pairs with which attribute (Dai et al., 2024). In multimodal models, analogous identifiers appear in VLMs: distinct binding codes are attached to an object's image tokens and its textual mentions, yielding cross-modal referential consistency (Saravanan et al., 2025). Complementary evidence points to *symbolic indexing* in VLMs, where attention heads compute content-independent spatial indices and later retrieve attributes by these indices, thereby implementing object-centric binding across modalities (Assouel et al., 2025). These findings highlight the role of hidden symbolic variables in reasoning. While prior analyses primarily examine existing identifiers, we show that simple input-augmented artifacts can *elicit* partition-specific identifiers that are linked across modalities and enhance grounding through tighter cross-modal attention and alignment.

**LVLM reasoning.** Efforts to improve reasoning in LVLMs have followed two main directions. Prompt-focused methods such as chain-of-thought prompting, majority voting, and test-time compute scaling aim to enhance reasoning without altering inputs. However, evaluations on multimodal reasoning benchmarks show that these approaches remain ineffective for tasks requiring visual or spatial reasoning, even when scaled substantially (Hao et al., 2025). In contrast, recent studies demonstrate that external scaffolding—adding annotations, grids, or partitions to the input image—consistently improves model performance across counting, spatial relations, and description tasks (Rudman et al., 2025; Izadi et al., 2025). While such findings establish the effectiveness of scaffolding, the mechanisms underlying these improvements remain poorly understood.

**Hallucination reduction.** Hallucination remains a central challenge for LVLMs, mainly arising from the decline of cross-modal attention during long text generation and over-reliance on language priors. Several inference-time methods have been proposed, including VCD (Leng et al., 2024), which down-weights tokens favored under distorted inputs; OPERA (Huang et al., 2024), which applies an over-trust penalty and retrospection allocation to rebalance attention; and SPARC (Jung et al., 2025), which progressively recalibrates visual attention to maintain relevance in long captions. Our work complements these approaches by identifying how visual scaffolding induces Grounding IDs that naturally sustain cross-modal binding, thereby reducing hallucination without additional inference tools.

## 7 CONCLUSION

In this work, we introduce a conceptual framework to explain how multimodal aligned external cues induce abstract *Grounding IDs* across related partitions. Through attention and embedding alignment analysis, we showed that these identifiers propagate across modalities, supporting partition-specific grounding and reducing the modality gap. Activation patching interventions confirmed that the associations are mediated by abstract identifiers rather than local features. Empirical evaluations further demonstrated that Grounding IDs enhance cross-modal attention, yield more faithful image descriptions, and reduce hallucinations. Since the approach relies only on simple, content-independent structures, it provides a model-agnostic strategy applicable to a wide range of tasks and models, including closed-source LVLMs.

Beyond empirical contributions, our methodology provides insights into mechanistic interpretability in multimodal models through both observational and causal tools. In addition, this study opens several directions for future research. One direction involves discovering circuits that support system-2 reasoning tasks, such as counting and spatial reasoning. Another arises from a key observation in this study: external cues in VLM inputs can reinforce the model's inherent grounding capability. Building on this, future work could explore integrating such cues during RL finetuning to further strengthen the model's internal ability to perform sequential scanning based on provided cues.

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

# A APPENDIX

## A.1 VISUAL TRAVERSAL AND POSITIONAL INDUCTIVE BIAS

To analyze how the model traverses the visual input, we compute mean cross-attention maps from textual tokens to image rows. These maps are averaged across 100 samples, respectively, and aggregated over layers 14–28. Each column corresponds to a decoding step, while rows are grouped top-to-bottom. This setup allows us to observe how attention flows across the image during generation.

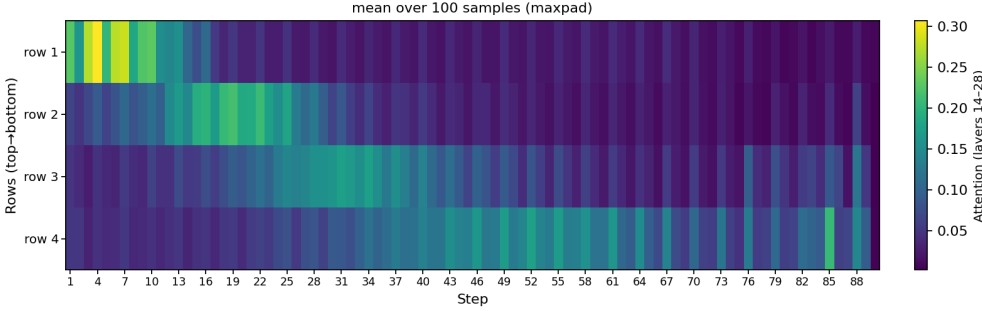

(a) Baseline input without structural cues (Averaged over 100 samples).

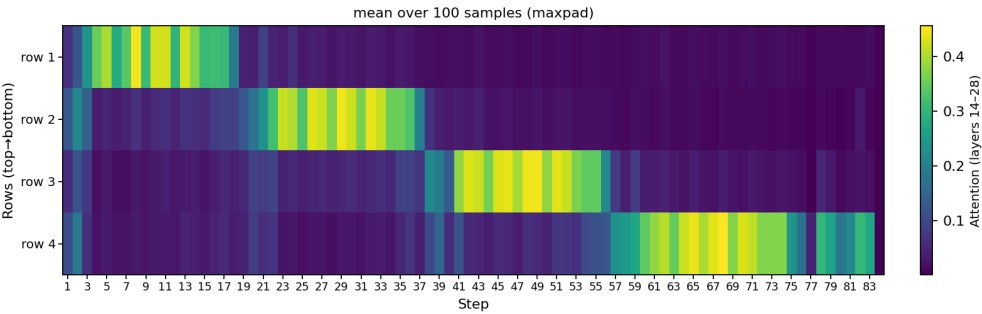

(b) Structured input with explicit row cues (Averaged over 100 samples).

Figure 9: Mean cross-attention maps (layers 14–28) from textual tokens to image rows. Structural cues amplify the model's inherent top-to-bottom, left-to-right traversal bias and produce sharper row-aligned attention patterns.

Fig. 9 reveals two consistent patterns:

1. **Inherent top-to-bottom, left-to-right traversal.** Even without explicit guidance, the model exhibits a natural reading-like behavior: starting from the upper-left region of the image and progressively shifting attention downward. This reflects the *positional encoding inductive bias* of the vision tokenizer, which encourages sequential exploration and may facilitate grounding by providing a consistent spatial reference across tokens.

2. **Effect of structural cues.** When explicit structure is added to the image (e.g., row separators or symbolic markers), the attention becomes sharper and more aligned with the intended reading order. Instead of diffuse activation, the model follows a clear row-by-row progression from top to bottom. This demonstrates how structural cues can *reinforce inherent grounding*, leading to more consistent visual traversal.

As a qualitative example, Figure 10 illustrates cross-attention heatmaps for individual objects. Each panel corresponds to a specific shape–color pair, showing how the model localizes and attends to the correct region during decoding. This highlights the row-by-row traversal pattern on a per-object basis.

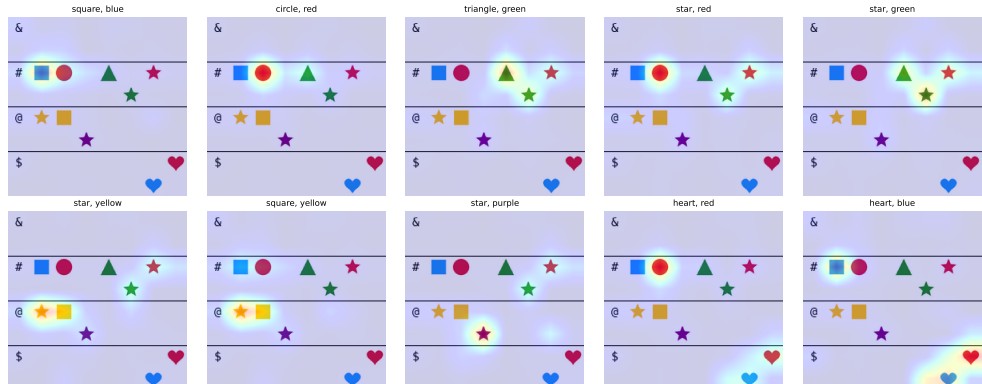

Figure 10: Examples of per-object traversal heatmaps for a structured image. Each panel shows the attention distribution for a specific shape–color pair. Structural cues guide the model toward sharper localization and promote systematic top-to-bottom and left-to-right traversal.

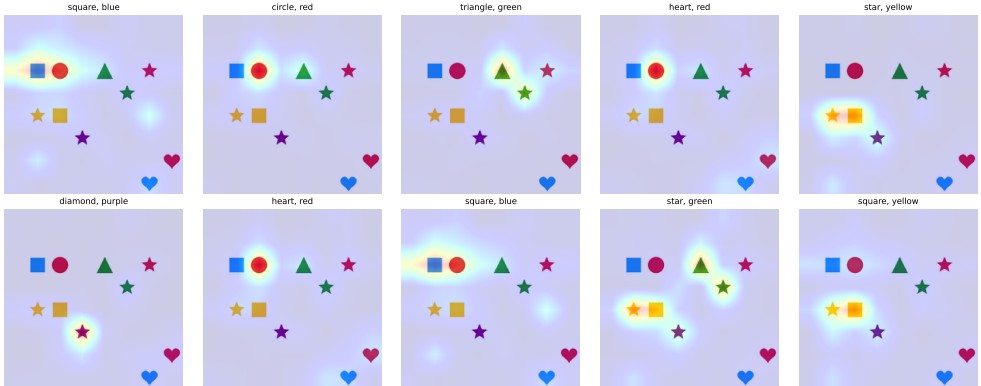

Figure 11: Examples of per-object traversal heatmaps for the baseline. Each panel shows the attention distribution for a specific shape–color pair. The baseline also exhibits a general top-to-bottom and left-to-right pattern, but accuracy degrades over generation.

Overall, these results indicate that the model does not scan the visual field randomly. Instead, it exhibits a systematic traversal strategy that can be strengthened by structural modifications to the input image, yielding more interpretable and faithful cross-modal grounding.

## A.2  LOGIT LENS ANALYSIS AND EVIDENCE FOR PROPAGATION

To better understand how symbolic cues influence within-partition binding, we apply a *logit lens* analysis to both raw and structurally manipulated images from our synthetic dataset containing seven distinct shapes and five colors. We project patch hidden states into the vocabulary space via the unembedding matrix, applying a softmax over the four symbol tokens (&, #, @, $). This measures how strongly each patch activates symbols, revealing the diffusion of symbolic evidence.

**Symbol vs Base Probabilities Grid**

Symbol

| & | $ 0.48 | & 0.93 | & 0.88 | & 0.85 | & 0.89 | & 0.80 | & 0.83 | & 0.93 |
| | & 0.90 | & 0.90 | & 0.83 | & 0.81 | & 0.82 | & 0.02 | & 0.81 | & 0.70 |
| # | & 0.99 | & 0.80 | # 0.59 | # 0.47 | & 0.49 | # 0.55 | & 0.46 | # 0.50 |
| | & 0.73 | & 0.48 | # 0.51 | & 0.48 | # 0.49 | # 0.51 | # 0.44 | & 0.49 |
| @ | # 0.49 | @ 0.57 | @ 0.41 | @ 0.34 | @ 0.38 | @ 0.35 | & 0.35 | @ 0.57 |
| | @ 0.67 | @ 0.53 | @ 0.45 | @ 0.35 | & 0.38 | @ 0.40 | @ 0.39 | @ 0.37 |
| $ | # 0.30 | $ 0.63 | $ 0.44 | $ 0.40 | $ 0.34 | $ 0.40 | $ 0.44 | $ 0.57 |
| | $ 0.65 | $ 0.35 | $ 0.44 | & 0.31 | $ 0.35 | $ 0.41 | $ 0.48 | # 0.30 |

Base

| $ 0.36 | $ 0.73 | $ 0.52 | $ 0.52 | $ 0.56 | $ 0.51 | $ 0.47 | $ 0.49 |
| # 0.63 | $ 0.44 | $ 0.38 | $ 0.34 | $ 0.36 | $ 0.37 | $ 0.44 | # 0.47 |
| # 0.41 | # 0.36 | # 0.34 | # 0.33 | # 0.33 | # 0.39 | $ 0.33 | & 0.33 |
| # 0.40 | # 0.34 | # 0.35 | & 0.36 | & 0.37 | # 0.33 | & 0.35 | & 0.35 |
| $ 0.31 | & 0.38 | & 0.35 | & 0.38 | & 0.42 | # 0.38 | & 0.40 | & 0.35 |
| # 0.49 | & 0.35 | # 0.36 | & 0.40 | & 0.34 | # 0.35 | & 0.32 | # 0.48 |
| # 0.32 | # 0.37 | # 0.40 | & 0.41 | & 0.33 | # 0.35 | # 0.33 | # 0.34 |
| $ 0.34 | & 0.40 | # 0.37 | & 0.40 | # 0.31 | # 0.41 | # 0.36 | # 0.31 |

Figure 12: Averaged logit lens predictions in the last layer. In the structured condition, row symbols (&, #, @, $) generate strong localized activations that propagate across the entire row, providing row-level binding. As expected, the baseline shows a random, unstructured pattern. Results are based on our synthetic dataset with seven shapes and five colors.

As shown in Figure 12, the manipulated images (left) produce sharper symbol activations that extend beyond their immediate positions, spreading across all patches within the corresponding row. This demonstrates how symbolic structure strengthens local grounding while also enabling coherent row-level propagation. By comparison, the baseline condition (right) lacks this structured diffusion, resulting in more diffuse and inconsistent attention patterns. Together, these findings highlight the role of explicit symbols in amplifying the model's evidence propagation and improving cross-modal grounding.

## A.3 Intra-Visual Attention Patterns

A key signal of grounding is whether patches from the same row interact more strongly, indicating that the model encodes rows as coherent positional units rather than arbitrary patch collections. To probe this effect, we compute pairwise cosine similarities between final-layer patch embeddings and compare the average similarity of patches within the same row to that of patches drawn from different rows. A clear gap between within-row and across-row similarity demonstrates that structural cues encourage row-wise grouping, effectively injecting an implicit positional prior into the representation space.

We further quantify this phenomenon with the **intra–inter cluster grounding (ICG)** score, defined as the difference between the mean within-row similarity and the mean across-row similarity:

$$ \text{ICG} = \frac{1}{R} \sum_{r=1}^{R} \text{Sim}_{\text{intra}}(r) \; - \; \frac{1}{R(R-1)} \sum_{r \neq r'} \text{Sim}_{\text{inter}}(r, r'), \tag{3} $$

where $R$ is the number of rows, $\text{Sim}_{\text{intra}}(r)$ is the mean cosine similarity between patches within row $r$, and $\text{Sim}_{\text{inter}}(r, r')$ is the average similarity between patches across rows $r$ and $r'$. A larger ICG score indicates stronger row-wise clustering and hence a clearer positional alignment in the image representation. Figure 13 plots the ICG score across layers for both the baseline and the structured setting. We observe that the structured condition (symbol-augmented inputs) consistently achieves higher ICG values than the baseline, indicating that structural cues enhance row-level grouping.

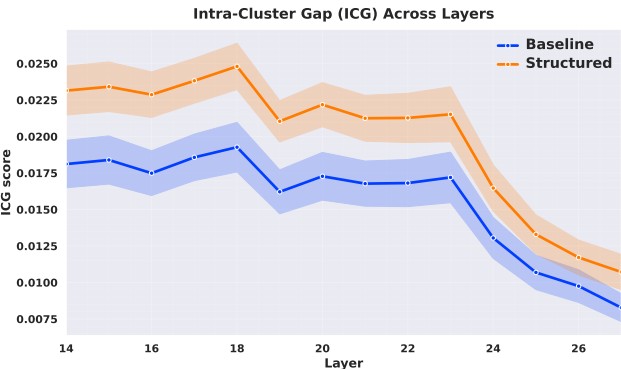

Figure 13: ICG scores across layers for baseline vs. structured inputs. Structured inputs consistently yield higher ICG, reflecting stronger row-level grouping.

## A.4 EVALUATION ON SYNTHETIC VARIANTS

We evaluate a range of structural variants of the synthetic dataset (7 shapes × 5 colors, with 20 objects per image) to understand how explicit row-level cues affect grounding. Each row in Table 4 reports **precision**, **recall**, **F1**, and **accuracy**.

The baseline condition provides no additional cues and yields weak performance. Introducing explicit structure consistently improves results, with the following variants:

- **Numbers (no line)**: Each row is tagged with a numeric index, but no horizontal separators are drawn. This already improves both precision and accuracy compared to the baseline.

- **Letters + Line**: Rows are separated by horizontal lines and tagged with distinct letter labels. This variant yields a significant boost in precision, indicating clearer disambiguation between rows.

- **Numbers + Line**: Rows are separated by lines and also numbered. This combination achieves the highest recall among all variants, suggesting that explicit row boundaries help the model correctly cover more objects.

- **Symbols + Line**: Each row is separated by a line and prefixed with a unique symbol (&, #, @, $). This provides the strongest positional anchor, achieving the best overall F1 and accuracy, reflecting both balanced precision and recall.

- **Grid 4×4**: A baseline layout where objects are arranged in a $4 \times 4$ grid. Two variants were tested: numbered with grid lines, and numbered without grid lines. Performance remained weaker than in the structured row version.

- **Numbers Right-Aligned + Line**: A variant of Numbers + Line where the numeric label appears at the end of the row rather than the beginning. While recall improves, the overall precision drops slightly, indicating less consistent alignment.

- **Symbols (no line)**: Each row is tagged with a unique symbol (&, #, @, $), but no horizontal separators are drawn. This already improves both precision and accuracy compared to the baseline.

- **Object Bounding (input-aware)**: Each object is enclosed within an explicit bounding box. While this provides a localized spatial cue, it is less effective than row-level structuring, resulting in only moderate improvements over the baseline.

- **Symbols on Objects (input-aware)**: Symbols are assigned at the object level instead of the row level. While this introduces additional visual markers, the lack of global row structure leads to reduced performance compared to row-based variants.

| Variant | Precision | Recall | F1 | Accuracy |
|---|---|---|---|---|
| Baseline | 0.14 | 0.45 | 0.21 | 0.12 |
| Numbers (no line) | 0.53 | 0.43 | 0.47 | 0.31 |
| Letters + Line | 0.64 | 0.48 | 0.55 | 0.38 |
| Numbers + Line | 0.56 | 0.58 | 0.57 | **0.40** |
| Symbols + Line | **0.65** | **0.59** | **0.62** | **0.40** |
| Grid 4×4 Numbered | 0.33 | 0.40 | 0.36 | 0.22 |
| Grid 4×4 Numbered (no lines) | 0.33 | 0.35 | 0.34 | 0.20 |
| Numbers Right-Aligned + Line | 0.39 | 0.55 | 0.46 | 0.30 |
| Symbols (no line) | 0.48 | 0.50 | 0.49 | 0.33 |
| Object Bounding Box | 0.43 | 0.48 | 0.44 | 0.32 |
| Symbols on Objects | 0.30 | 0.38 | 0.29 | 0.19 |

Table 4: Performance metrics (precision, recall, F1, accuracy) on the scene description task across synthetic variants with 20 objects. Explicit structural cues (such as lines, labels, or symbols) consistently improve performance compared to the baseline.

### A.5 IMPLEMENTATION DETAILS AND QUALITATIVE EXAMPLE

We employed two types of prompts for instruction tuning and evaluation: a *structured prompt* that explicitly encodes row-level layout with symbols, and a simpler *baseline prompt* without structural cues.

**Structured Prompt.**

> **Task:** You are presented with an image containing multiple objects. Your task is to identify all objects with color in the image. Each object must be exactly `<color> <shape>`. Scan the image sequentially based on horizontal lines and rows that exist in the image. There are exactly 4 horizontal rows in the image.
>
> Use *only* these shapes (exact strings): `square, circle, triangle, diamond, star, moon, heart`
>
> Use *only* these colors (exact strings): `red, green, blue, yellow, purple`
>
> Important: If the same object appears multiple times in a row, repeat its label for each instance (e.g., `blue square, blue square, red circle`).
>
> **Output format (print *exactly* this structure, no extra text):**
>
> ```
> Row &: <label>, <label>, ...
> Row #: <label>, <label>, ...
> Row @: <label>, <label>, ...
> Row $: <label>, <label>, ...
> ```

**Baseline Prompt.**

> **Task:** You are presented with an image containing multiple colored objects. Your task is to identify all objects in the image and report each as `<color> <shape>`.
>
> Use *only* these shapes (exact strings): `square, circle, triangle, diamond, star, moon, heart`
>
> Use *only* these colors (exact strings): `red, green, blue, yellow, purple`
>
> If the same object appears multiple times in a row, repeat its label for each instance (e.g., `blue square, blue square, red circle`).
>
> **Output format (print *exactly* this structure, no extra text):**

```
        Output a single line: a comma-and-space separated list of labels,
        where each label is "<color> <shape>".
```

**Structured Prompt Output:**

```
Row &: purple moon, purple star, blue moon
Row #: yellow heart, blue star, purple diamond
Row @: blue circle, purple circle
Row $: yellow triangle, blue heart
```

**Baseline Prompt Output:**

```
purple moon, purple star, purple diamond, yellow heart,
blue circle, blue square, blue circle, blue circle,
blue circle, blue circle, yellow triangle, blue heart
```

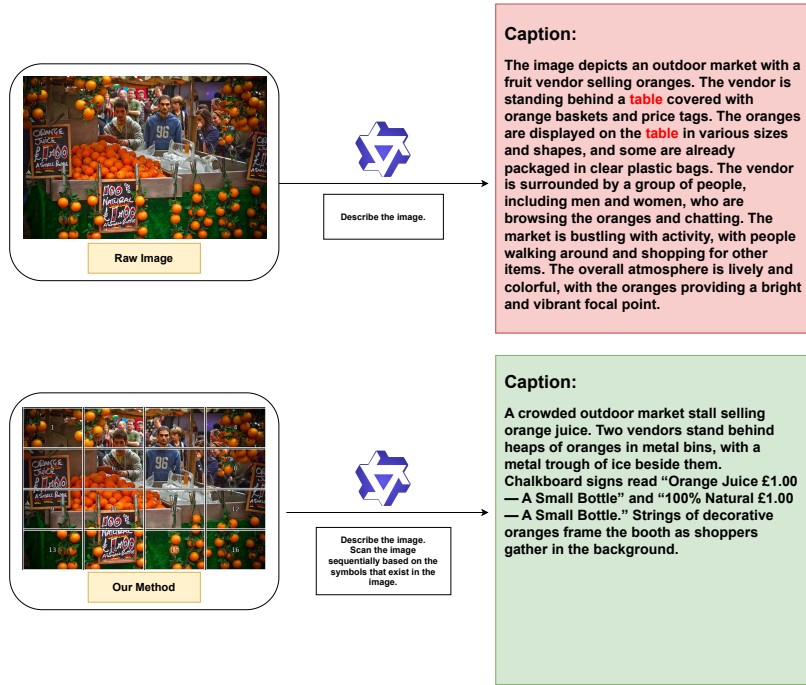

Figure 14: Qualitative example comparing captions generated with and without structured scaffolding. The figure shows the input image, the applied scaffolding structure, the caption produced by our method, and the corresponding baseline caption. Hallucinated objects in the baseline output are highlighted in red.

## A.6 EVALUATION BENCHMARKS FOR OBJECT HALLUCINATION

Object hallucination is a key failure mode of LVLMs, occurring when models generate content that is unsupported by the visual input. To quantify this phenomenon, we adopt two complementary evaluation frameworks: **CHAIR** and **POPE**. While both focus on grounding and image relevance, they differ in format, granularity, and the types of hallucination they expose.

### A.6.1 CHAIR: CAPTION HALLUCINATION ASSESSMENT WITH IMAGE RELEVANCE

The *CHAIR* metric evaluates whether objects mentioned in generated captions are actually present in the image. Using ground-truth segmentations and reference captions from the MSCOCO dataset,

CHAIR measures hallucination at two levels:

$$\text{CHAIR}_i = \frac{|\{\text{hallucinated objects}\}|}{|\{\text{all objects mentioned}\}|}, \tag{4}$$

$$\text{CHAIR}_s = \frac{|\{\text{sentences with hallucinated objects}\}|}{|\{\text{all sentences}\}|}. \tag{5}$$

Here, $\text{CHAIR}_i$ quantifies hallucination at the object-instance level, while $\text{CHAIR}_s$ captures the proportion of captions containing hallucinated objects. Lower scores indicate better grounding and fewer hallucinated mentions.

We compute CHAIR scores on 500 randomly selected images from the MSCOCO validation set, using model-generated captions and official COCO object annotations as references.

### A.6.2 POPE: PROBING OBJECT-PRESENCE HALLUCINATION

The *POPE* benchmark probes whether models falsely predict the presence of objects not found in the image. Unlike CHAIR, which evaluates free-form captioning, POPE uses templated binary questions of the form: "*Is there a {object} in the image?*" and compares the model's answer to COCO annotations. The focus is on identifying *false positives*, i.e., cases where the model incorrectly affirms the presence of objects that are absent.

POPE defines three targeted subsets to assess different sources of hallucination:

- **Random**: Object categories are randomly sampled and paired with images in which the object is absent, probing baseline hallucination without context.
- **Adversarial**: Object prompts are selected to be semantically plausible within the scene (e.g., "fork" in a kitchen) but are not present, testing model susceptibility to co-occurrence priors.
- **Popular**: High-frequency object categories are paired with unrelated images to measure overprediction due to training frequency bias.

To ensure comparability with CHAIR, we evaluate POPE using questions corresponding to the same 500 randomly selected images from the MSCOCO validation set. For each model response, we compute standard classification metrics: **precision**, **recall**, **F1 score**, and **accuracy**, defined as follows:

$$\text{Precision} = \frac{\text{TP}}{\text{TP} + \text{FP}}, \qquad\qquad \text{Recall} = \frac{\text{TP}}{\text{TP} + \text{FN}}, \tag{6}$$

$$\text{F1 Score} = \frac{2 \cdot \text{Precision} \cdot \text{Recall}}{\text{Precision} + \text{Recall}}, \qquad \text{Accuracy} = \frac{\text{TP} + \text{TN}}{\text{TP} + \text{FP} + \text{TN} + \text{FN}}, \tag{7}$$

where TP, FP, TN, and FN denote true/false positives and negatives. These metrics together capture hallucination behavior, balancing precision (avoiding false affirmations) with recall and accuracy (correctly rejecting absent objects).

### A.6.3 EVALUATION ON POPE BENCHMARK

In addition to CHAIR, we evaluate our structured scaffolding method using the POPE benchmark, which probes object hallucination through structured binary questions. As described in Appendix A.6.2, POPE defines three subsets—*Random*, *Adversarial*, and *Popular*—each targeting different hallucination biases such as language priors, scene co-occurrence, and object frequency.

While our proposed strategy is designed to strengthen grounding during *long-form generation*, we also evaluate on POPE to ensure coverage across widely used hallucination benchmarks. Although POPE is less aligned with our method's strengths, results remain robust and comparable, demonstrating that the approach generalizes beyond descriptive settings.

We use the same 500 MSCOCO validation images as in the CHAIR evaluation. From the POPE benchmark, we select yes/no question subsets corresponding to these images. Each image has six questions per subset (Random, Adversarial, Popular), yielding **3,000 questions per subset** and

**9,000 in total** per model. We evaluate both baseline and structured versions of LLaVA-1.5 and Qwen2.5-VL using standard binary classification metrics: precision, recall, F1 score, and accuracy.

Table 5: Results of POPE evaluation on the MSCOCO dataset across Random, Popular, and Adversarial splits. Metrics reported are Accuracy, Precision, Recall, F1 Score, and Yes (%) for LLaVA-1.5.

| Model | POPE | Method | Accuracy | Precision | Recall | F1 Score | Yes (%) |
|---|---|---|---|---|---|---|---|
| LLaVA-1.5 | Random | Base | 87.27 | 97.47 | 76.66 | 85.82 | 39.53 |
| | | OPERA | 86.17 | 97.00 | 74.88 | 84.52 | 38.93 |
| | | VCD | 85.43 | 93.86 | 75.99 | 83.99 | 40.70 |
| | | SPARC | 85.10 | **98.90** | 71.34 | 82.89 | 36.50 |
| | | **Structured** | **89.23** | 95.19 | **82.75** | **88.53** | 43.67 |
| | Popular | Base | 85.93 | 94.28 | 76.63 | 84.54 | 40.80 |
| | | OPERA | 84.93 | 93.99 | 74.78 | 83.30 | 39.97 |
| | | VCD | 83.67 | 89.89 | 76.06 | 82.40 | 42.53 |
| | | SPARC | 83.67 | **97.06** | 71.74 | 82.50 | 39.67 |
| | | **Structured** | **86.07** | 88.77 | **82.77** | **85.67** | 46.90 |
| | Adversarial | Base | 81.43 | 86.17 | 75.15 | 80.28 | 43.87 |
| | | OPERA | 82.07 | 87.76 | 74.88 | 80.81 | 43.03 |
| | | VCD | 80.57 | 83.94 | 76.01 | 79.78 | 45.67 |
| | | SPARC | **84.93** | **98.90** | 70.98 | 82.64 | 36.27 |
| | | **Structured** | 83.50 | 89.03 | **76.72** | **82.42** | 43.43 |

Table 6: Results of POPE evaluation on the MSCOCO dataset across Random, Popular, and Adversarial splits. Metrics reported are Accuracy, Precision, Recall, F1 Score, and Yes (%) for Qwen2.5-VL.

| Model | POPE | Method | Accuracy | Precision | Recall | F1 Score | Yes (%) |
|---|---|---|---|---|---|---|---|
| Qwen2.5-VL | Random | Base | 89.73 | 98.39 | 80.91 | 88.80 | 41.37 |
| | | OPERA | 88.83 | **98.76** | 78.85 | 87.69 | 40.27 |
| | | VCD | **89.90** | 98.55 | **81.13** | **88.99** | 41.43 |
| | | SPARC | 85.33 | 95.91 | 71.76 | 82.10 | 35.07 |
| | | **Structured** | 85.67 | 98.22 | 72.90 | 83.69 | 37.43 |
| | Popular | Base | 88.83 | 96.07 | 81.09 | 87.95 | 42.40 |
| | | OPERA | 87.40 | 96.70 | 77.63 | 86.12 | 40.43 |
| | | VCD | **89.00** | 96.09 | **81.44** | **88.16** | 42.63 |
| | | SPARC | 84.77 | 96.88 | 72.02 | 82.62 | 37.37 |
| | | **Structured** | 85.07 | **96.92** | 72.72 | 83.09 | 37.87 |
| | Adversarial | Base | **87.43** | 93.16 | 81.02 | 86.66 | 43.83 |
| | | OPERA | 86.43 | **94.35** | 77.62 | 85.17 | 41.30 |
| | | VCD | 87.40 | 92.64 | 81.49 | **86.71** | 44.37 |
| | | SPARC | 81.17 | 80.35 | **82.75** | 81.53 | 51.73 |
| | | **Structured** | 83.73 | 93.60 | 72.65 | 81.80 | 39.07 |

Importantly, we observe that the structured version achieves lower **recall** than the baseline. Analyzing the false negatives reveals systematic patterns, such as objects positioned across grid boundaries or large objects spanning multiple cells. In this work we adopt a general prompting strategy, but more tailored prompts that explicitly define such corner cases could further reduce false negatives. Additionally, techniques such as ensembling over diverse structural partitions may help compensate for the limitations of any single structure.

## A.7 HYPERPARAMETERS FOR HALLUCINATION-MITIGATION METHODS

For all hallucination experiments on MSCOCO (CHAIR and POPE), we use a unified decoding configuration unless otherwise specified: greedy decoding with temperature $T = 0.0$, top-$p = 1.0$, beam size 1, and a maximum of 512 new tokens.

Below we list the exact hyperparameters used for each hallucination-mitigation baseline, following the recommendations of the original papers.

**VCD (Visual Contrastive Decoding).**   We follow the hyperparameters suggested in the VCD paper. The decoding coefficients are fixed to:

$$\alpha = 1, \qquad \beta = 0.1, \qquad \gamma = 0.1.$$

For the diffusion-based distortion process, we adopt the recommended number of noise steps, $T = 500$. These settings are used for both LLaVA-1.5 and Qwen2.5-VL.

**OPERA.**   We use the default configuration recommended in the OPERA paper. The method introduces an over-trust logit penalty and a retrospection-allocation mechanism, with parameters:

$$N_{\text{can}} = 5, \qquad \sigma = 50, \qquad \alpha = 1, \qquad r = 15.$$

These unified settings are applied across all evaluated models, including LLaVA-1.5 and Qwen2.5-VL. We use beam size 5 following the authors' implementation.

**SPARC.**   For SPARC, we follow the recommended hyperparameters from the original paper. Global parameters include:

$$\alpha = 1.1, \qquad \beta = 0.1.$$

Attention recalibration is applied across all layers. The token-selection threshold $\tau$ and the attention-extraction layer depend on the model:

- **LLaVA-1.5:** $\tau = 1.5$, selected layer $= 20$.
- **Qwen2.5-VL:** $\tau = 3.0$, selected layer $= 18$.

A.8   ACTIVATION SWAPPING (INTERCHANGE INTERVENTION)

**Data generation.**   We generate 100 synthetic scenes ("samples"). Each scene is a four-row grid with exactly one *symbol* and one *object* per row. We draw four *distinct* shapes (e.g., circle, diamond, square, triangle) and eight *distinct* colors (chosen without replacement from a nine-color palette) and assign one unique (shape, color) pair to each row. Symbols are randomly permuted over the physical rows (top→bottom) in every sample.

**Pairing and target selection.**   For each sample $x$, we select a paired sample $x'$ and two distinct target symbols. Because each scene contains four unique (shape, color) pairs, targets are unambiguous within a scene. To isolate intervention effects, we impose a *symmetric mismatch* between $x$ and $x'$: for each chosen symbol $s$, the associated object in $x$ and the associated object in $x'$ must differ in *both* shape and color (i.e., $\text{shape}_x(s) \neq \text{shape}_{x'}(s)$ and $\text{color}_x(s) \neq \text{color}_{x'}(s)$). This design also rules out the *symbol-activation* alternative hypothesis—namely, that the model might answer by reading object attributes cached in the symbol's own representation. Because our swaps patch **only object image tokens** with a one-token dilation (**pad = 1**) and *never* include symbol tokens, and because paired objects differ on both attributes, any post-swap answer that follows the injected attributes cannot be explained by information stored in symbol activations.

**Intervention.**   For each pair $(x, x')$: (i) we first query both contexts *without intervention* to obtain baseline answers of the form "What is the *shape* / *color* of the object associated with symbol $S$?"; (ii) we then perform an *object-only activation swap*. Let $\mathcal{I}_r$ be the set of *image-token* indices belonging to the **object** in the referenced row $r$ (obtained by mapping patch centers to the object's bounding box); we dilate this mask by one token in each axis to get $\tilde{\mathcal{I}}_r$ (**pad=1**). We swap the hidden activations at $\tilde{\mathcal{I}}_r$ between the two runs and leave all other positions *unchanged*—in particular, we do *not* alter (a) any **text tokens** that mention symbols, or (b) image tokens occupied by the **symbol glyphs** themselves. (iii) We re-ask the same questions and record post-swap answers for the two targets.

**Outputs.**   For each context we log pre-/post-swap answers and token-level logits for the queried attributes.

**Prompt.** For each target symbol, we use the same wording across both contexts and substitute `<row_symbol>` with one of &, $, #, @:

```
    Scan the image using the symbols on the left (&, $, #, @) as
    row labels.
    What is the shape of the object in the ``<row_symbol>'' row?
```

(For color queries, replace "shape" with "color".)

### A.8.1 DISJOINT-SYMBOL SWAP (NO-VISUAL-CUE PROMPT)

To further assess the role of abstract identifiers, we repeat the activation-swap experiment with *disjoint symbol sets* across contexts. The source image uses $\{\&,\$,\#,@\}$, while the target image uses $\{+, \times, \%, !\}$, with no overlap (Fig. 6b). After object-only activation patching from the source into the target (pad $= 1$; symbol tokens never swapped), we *intentionally query the target* using a **source** symbol (e.g., &), which is not physically present in the target image. The query omits any enumeration of the available symbols so as not to contradict the target's visual glyphs.

**Prompt.** We use the same wording as above, but without listing the four symbols. For a chosen `<row_symbol>` (e.g., &) we ask:

```
    Scan the image using the symbols on the left as row labels.
    What is the shape of the object in the ``<row_symbol>'' row?
    If there is no object corresponding to ``<row_symbol>'' in
    the image, answer none.
```

(For color queries, replace "shape" with "color".)

**Bias control and evaluation protocol.** Allowing the response *none* mitigates bias toward hallucinating an answer when the queried symbol is not present. For evaluation, the accuracy reported in the paper is computed on the subset of examples where, in the *baseline* (non-intervened) condition, the model correctly answered *none* for the queried symbol. This isolates the effect of the intervention from cases where the model would have produced a spurious non-none answer even without patching.

### A.9 EXTENDED EXPERIMENTS ON ALTERNATIVE MODELS

Experiments in Sections 3 and 4 were originally conducted on Qwen2.5-VL (7B). Here, we present additional experiments on models with different sizes and architectures. Figures 15 and 16 show within- and cross-modality attention analyses for InternVL3.5 (8B) and Qwen2.5-VL (3B), which exhibit the same attention progression as Qwen2.5-VL (7B) (Figure 2).

We also repeated the activation-swap experiment from Section 4 on InternVL3.5 (8B) and Qwen2.5-VL (3B). The models exhibit the same qualitative behavior as Qwen2.5-VL (7B). For InternVL3.5, swap accuracy is 0.75 for shape and 0.82 for color, well above the random chance level of 0.25 for shape and 0.125 for color. For Qwen2.5-VL (3B), swap accuracy is 0.93 for shape and 0.73 for color, indicating a consistent dependence on Grounding IDs. Figure 17 shows the corresponding log-prob diagrams, which follow the same pattern observed in Qwen2.5-VL (7B) (Figure 4b).

Finally, we repeat the hallucination mitigation experiments on MS-COCO using two additional open-source models of different sizes: Qwen2.5-VL (3B) and InternVL3.5 (8B). Results are reported in Table 7.

### A.10 CLARIFICATION OF TERMINOLOGY

Several terms used in the main paper relate to mechanisms that have been discussed in prior work on internal representations in language and vision–language models. For clarity, we provide brief descriptions here.

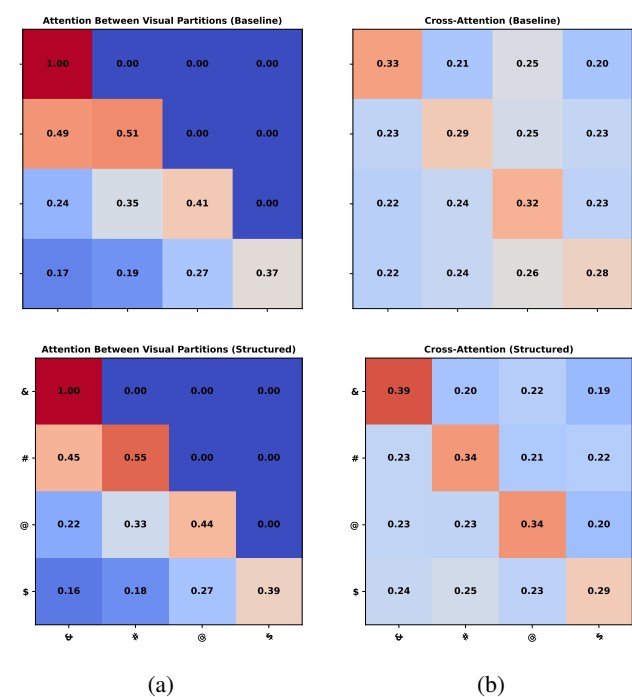

Figure 15: Attention patterns of InternVL3.5 (8B) under baseline (top) and structured (bottom) inputs for the scene description task. (a) Within-modality visual attention. (b) Cross-modality attention.

| Model | Method | CHAIR$_s$ ↓ | CHAIR$_i$ ↓ |
|---|---|---|---|
| Qwen2.5-VL 3B | Baseline | 32.40 | 8.82 |
| | **Structured** | **24.40** | **7.32** |
| InternVL3.5 8B | Baseline | 32.40 | **6.09** |
| | **Structured** | **26.40** | 8.08 |

Table 7: CHAIR results on 500 MS-COCO samples for Qwen2.5-VL (3B) and InternVL3.5 (8B).

**Symbolic.** We use the term *symbolic* to refer to the fact that external cues (such as characters or markers) function as explicit and discrete symbols that index different regions of the input. They can be consistently referenced by the model both in text and in internal computation.

**Identifier.** An *identifier* denotes an internal code that the model assigns to all tokens associated with the same cue. This identifier allows the model to bind visual and textual elements referring to the same partition.

**Latent.** The identifier is *latent* because it does not appear explicitly in the visible input. While the external cue is visible, the identifier itself arises in the model's hidden activations and can be detected through probing and representation analysis.

**Vector.** We refer to the identifier as a *vector* because it is expressed as a direction or component in activation space rather than as a discrete token.

**Abstract.** The identifier is *abstract* because it does not encode perceptual details such as color or shape. Instead, it represents the relational property of membership in a specific partition, independent of the visual content of that partition.

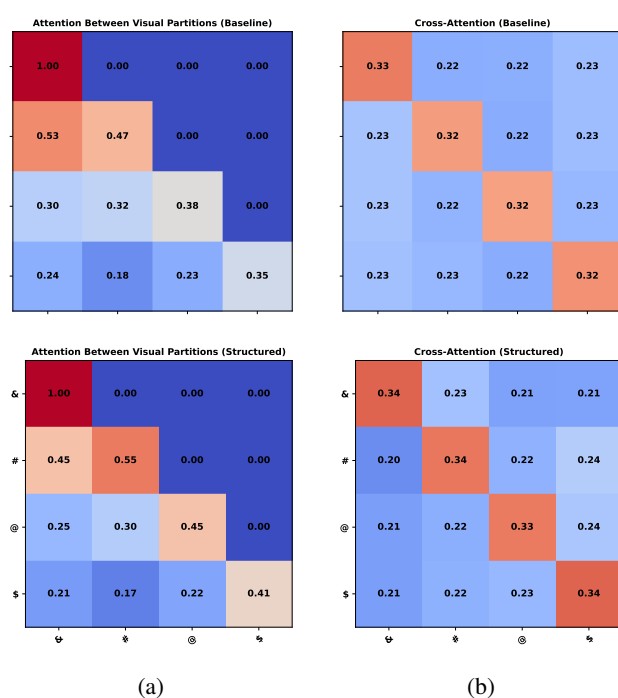

Figure 16: Attention patterns of Qwen2.5-VL (3B) under baseline (top) and structured (bottom) inputs for the scene description task. (a) Within-modality visual attention. (b) Cross-modality attention.

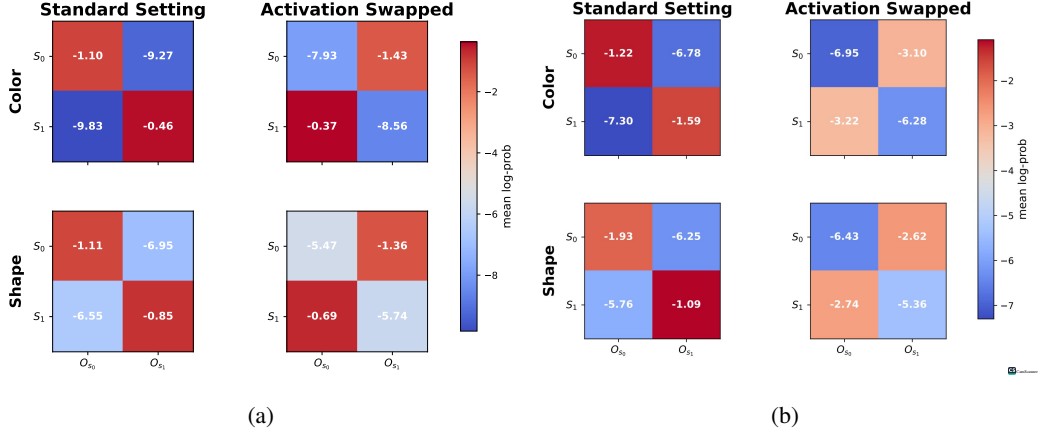

Figure 17: Average log probabilities of $c$ and $c^*$ over valid row–symbol–object combinations for InternVL3.5 (a) and Qwen2.5-VL 3B (b). Rows and columns correspond to the selected query symbols and their associated objects.

