# OpenReview forum: "Uncovering Grounding IDs: How External Cues Shape Multi-Modal Binding"
_ICLR.cc/2026/Conference — Submitted to ICLR 2026_

### Official Review · Reviewer_Gyen · 2025-10-27

**Soundness:** 3
**Presentation:** 3
**Contribution:** 3
**Rating:** 4
**Confidence:** 2

**Summary:**

This work introduces Grounding IDs to explain why simple symbolic cues, lines, or partitions added to an image enable large vision-language models to align visual and textual information more accurately and perform structured reasoning. Through representation alignment, activation swapping, and attention analysis, the authors demonstrate that these region-specific identifiers play a crucial role in cross-modal binding and can markedly reduce hallucinations, improving the factual consistency and interpretability of model outputs.

**Strengths:**

Conceptual novelty: Grounding IDs are introduced for the first time, offering a causally verifiable account of how external visual structures enhance multimodal reasoning and alignment.

Rigorous analysis: A systematic investigation—combining representation alignment, activation intervention, and attention diagnostics—demonstrates the mechanism’s validity.

High practical value: The approach markedly reduces hallucinations while incurring negligible computational overhead and remains broadly applicable and extensible.

Well-designed experiments: The study integrates controlled synthetic data with standard MS-COCO benchmarks, balancing theoretical exploration and real-world validation.

**Weaknesses:**

1. The writing of the paper is somewhat informal, and key terms such as “symbolic indexing”, “latent identifiers”, and “abstract vectors” are not clearly defined early on. This may reduce the clarity and conceptual coherence, making it more difficult for readers to fully understand the proposed framework.

2. The experiments mainly focus on horizontal lines and symbolic structures, limiting generalization. Exploring other cues, such as color partitions, semantic regions, or dynamic elements, could provide a more comprehensive evaluation.

3. The limited dataset sizes, with around 100 samples for synthetic experiments and 200 for hallucination evaluation, may restrict the statistical reliability and generalizability of the results.

4. The current evaluation focuses on a single model size, leaving open the question of whether Grounding IDs can generalize to smaller (1B–3B) or larger (30B+) LVLMs.

**Questions:**

Since the grid lines or symbols and the scanning prompt were always presented together, it remains unclear whether Grounding IDs are primarily triggered by the visual cues, the textual instruction, or the combination of both. A 2×2 orthogonal ablation (visual-only, prompt-only, both, neither) would help disentangle these factors and clarify whether they are complementary or redundant. Could the authors provide such control experiments or justify why they were not included?

---

> ### Author Response · Authors · 2025-11-23
>
> We appreciate the reviewer for the thoughtful assessment and for highlighting the conceptual novelty of Grounding IDs, the strength of our causal analyses, and the practical value of improved grounding and reduced hallucinations.
>
> > **W1**
>
>
> We acknowledge that several technical terms may be unfamiliar to a broader readership. These terms are commonly used in related work, and prior papers do not typically provide formal definitions. Due to space limitations, we also did not initially define them in the main text, but made this clear in the revised version. In this study:
> * **Symbolic** is used because the external cues act as explicit symbols that index different regions of the input, allowing the model to reference and organize them consistently. Our causal results also show that Grounding IDs behave as modular and transferable components rather than entangled distributed patterns.
> * **Identifier** is used because the model learns an internal code that uniquely tags all tokens associated with the same partition, supporting reliable binding between multimodal tokens.
> * **Latent** is used because this identifier is not present in the visible input (while its cause, the external cue, is visible in the input). It is formed internally and becomes measurable only in the hidden activations, especially in middle to late layers. Our causal experiments further show that these hidden identifiers can exert stronger influence than the visible cues adjacent to the objects.
> * **Vector** is used because the identifier is represented as a representational vectors in activation space rather than a discrete token.
> * **Abstract** is used because the identifier does not encode specific visual attributes such as color or shape, but instead represents the relational concept of “belongs to this partition,” independent of the specific input.
>
> &nbsp;
> ___
>
>
> > **W2**
>
>
> In this work, we focus on simple, input-agnostic structures because they are more general and do not require external tools, making them suitable for studying internal mechanisms. Our goal is to analyze how external cues influence model representations, rather than to exhaustively explore a broad range of cue designs. As shown in Section A.4 of the appendix, our analysis covers combinations of simple structures (lines and grids) and simple marks (letters, numbers, symbol characters).
>
>
> Extending the idea to dynamic or input-aware cues is also an interesting direction. For example, external cues could be based on semantic regions in agentic systems that use object detection tools. While this is beyond the scope of our main study, we have expanded Section A.4 in the revised version with two input-aware baselines (Object Bounding Box and Symbols on Objects) to show the potential applicability:
>
> |Variant|Prec|Rec|F1|Acc|
> |-|-:|-:|-:|-:|
> |Baseline| 0.18|0.46|0.25|0.15|
> |Object Bounding Box|0.43|0.48|0.44|0.32|
> |Symbols on Objects|0.30|0.38|0.29|0.19|
>
>
> &nbsp;
> ___
>
> > **W3 & W4**
>
> We agree that 100 samples per configuration may be limited. We updated the experiments with 500 samples in the revised version and observed small differences. Below, we show the entries for the baseline and structured approaches using 500 samples:
>
>
> * **Synthetic**:
>
> |#Objs|Method|Prec|Rec|F1|Acc|
> |-:|-|-:|-:|-:|-:|
> |10|Baseline|0.56|0.56|0.58|0.42 |
> |10|Structured|0.74|0.58|0.65|0.48|
> |15|Baseline|0.30|0.49|0.37|0.24|
> |15|Structured|0.67|0.53|0.59|0.46|
> |20|Baseline|0.14|0.45|0.21|0.12|
> |20|Structured|0.65|0.59|0.62|0.40|
>
>
>
> * **MS-COCO**, including one new small model **Qwen2.5-VL 3B**:
>
> |Model|Method|CH_s|CH_i|
> |-|-|-:|-:|
> |Qwen2.5-VL 3B|Base|32.4|8.82|
> |Qwen2.5-VL 3B|Structured|24.4|7.32 |
> |LLaVA-1.5|Base|51.6|13.20|
> |LLaVA-1.5|Structured|41.0|12.04|
> |Qwen2.5-VL|Base|32.4|7.97|
> |Qwen2.5-VL|Structured|27.2|5.36|
> |GPT-4o|Base|29.2| 6.40|
> |GPT-4o|Structured|23.2|5.81|
> |Gemini-2.5-Pro|Base|44.2|8.64|
> |Gemini-2.5-Pro|Structured|37.4|7.28 |
>
>
> While we already covered a range of **open-source and closed-source models** in our behavioral experiments, we also added a new appendix section (A.9) in the revised version that re-generates results from Sections 3 and 4 using two additional architectures, InternVL3.5 (8B) and Qwen2.5-VL (3B).
>
> &nbsp;
> ___
>
>
> > **Q1**
>
> We would like to clarify that the 2×2 orthogonal ablation (visual-only, prompt-only, both, neither) is already covered in the paper.
> One of the main findings of this work is that structural cues must be referenced in both modalities to be effective (see Lines 69 and 327).
> We also ablate this factor in Table 1 by including two unimodal baselines (text-only structure and image-only structure). In addition, Section A.4 of the appendix examines further combinations of structures.

---

### Official Review · Reviewer_Q3Bc · 2025-10-29

**Soundness:** 2
**Presentation:** 1
**Contribution:** 2
**Rating:** 4
**Confidence:** 3

**Summary:**

This method investigates the hypothesis that LVLMs associate "Grounding IDs" with certain partitions of data in different modalities, for example associating the symbol "&" with a certain sub-region of an image. This investigation is carried out with experimental testing of LVLMs in controlled situtations, in particular with images of coloured objects positioned in different regions of the image.

**Strengths:**

The hypothesis seems interesting and is investigated in a straight-forward and honest fashion. The question, ie what meaning do LVLMs give to abstract symbols which might describe image properties, seems important. The results seem convincing.

**Weaknesses:**

I found the paper difficult to understand, in particular Sections 3-4, not due to bad writing, but due to lack of precise details. I put specific examples of this below. Furthermore, I found the controlled experiments a bit limited. When the experiments went on to more real-world examples, using Qwen and LLaVA, I did not see where the notion of abstract symbols, which were hypothesised in the paper, intervened. It seems that you just add lines to the image, how does this show that these models actually use language symbols ?

However, I re-state that I found the work honest and to the point. Therefore, I would be willing to change my rating if the two following things were done:
- Significantly clear up and clarify sections 3-4, addressing the questions below
- Explain how the conclusions of sections 3-4 apply to the more real-world example with LlaVa and Qwen, in particular, how do you know that the abstract symbols are really used here (couldn't the model just look at the lines you have drawn ?).

**Questions:**

In Section 3, I do not understand which LVLM specifically is chosen here, it seems to be never specified. Are you using it pre-trained and frozen ? Is there any fine-tuning ?

The "baseline" is not explained in Section 3. It is very difficult to trust any results if we do not know what this baseline is, is it more simple or more complicated than the structured input ? Does it contain similar types of data ? Same shapes ? etc...

Section 4 is difficult to understand, again due to lack of details. I did not understand the process of swapping, what precisely was done. I got confused between swapping objects in an image and the swapping of activations inside the network. Which of these is done ? (one, both). What is "activation patching" ? You say "If we follow the standard setting without intervention ... the accuracy drops from 1.00 to 0.02". At this point I did not know what was being done exactly. If I understand correctly, you ask it to say what is in the row &, for example, but then switch the meaning of the symbols. So, the question is, was there any fine-tuning on the model? If so, then of course since you are forcing the model to associate this position with a symbol, and then when you carry out the inference, you change the meaning, so it will for sure not get it correct. If not, then I do not understand why it would give any meaning to that abstract symbol anyway, so we should not get such a change in accuracy. But since it is difficult to understand exactly what is done

- p. 3, line 118: "the model is tasked ..." setence repeated twice
- p. , line 122: "each token must be paired with ..." do you mean that the llm must do this ? Not super clear.
- p.3, line 124: what is a "regular expression" ? This may be because I am not as familiar with the domain as others
- p. 3, line 129: what is the "baseline" ? Not clear or stated as far as I can tell.
- p. 4, line 186: "rows are labelled ... to prevent sequential order cues". If there is fine-tuning or any sort of training, then the model can simply associate these symbols with numbers. But it is not clear if the model is fine-tuned or not
- p. 5, line 223: "host context". What is a host context ?
- p. 5, line 217-218: I found the sentences "swap their corresponding patch activations ... the object ... is inserted" quite difficult to follow. What does "the object in row $ of $c'$ is inserted into row ..." mean ? Do you mean that you swap the values of the activations (neurons) of the LVLM at some layer ? If so, which layer ? Before, after attention ? Does it have anything to do with attention, or just the plain activations (neurons) ? This is far too vague for me to be able to correctly understand.

---

> ### Author Response · Authors · 2025-11-23
>
> Thank you for the thoughtful and constructive review. We appreciate your positive assessment of the main hypothesis, the experimental design, and the relevance of studying how LVLMs interpret abstract symbols.
>
> ### Clarifying Sections 3 - 4:
>
>
> We are grateful for the detailed feedback regarding inadequate explanations in Sections 3-4. We acknowledge that several descriptions and interpretations are compact. This is because the paper presents the studied phenomenon from three angles (observational, causal and behavioral), and the space limit forced us to remove some explanations. With one additional page in the revised version, we provided sufficient clarifications.
>
>
> **Model in Section 3**:  Although the model is mentioned at Line 160 (Qwen2.5-VL), we agree that it should also be explicitly stated in the experimental setup paragraph of this section. We focus on enhancing inference-time reasoning, so the** models are frozen and no training or fine-tuning is applied**.
>
> **Baseline**: The baseline matches the notion used in VISER: the standard LVLM input (image plus prompt) for a visual reasoning task such as scene description. Examples of baseline versus structured inputs appear in Fig. 13 and Fig. 2a, and prompt templates are provided in Appendix A.5. We make this more clear at the beginning of Section 3 in the revised version.
>
> **Activation patching**: For activation patching, we transfer layer activations of selected tokens between two contexts (see Lines 216, 217, 201). Transferring objects in the raw input would not support causal analysis. Although the current text mentions activations several times (for example, Line 216) and cites prior work to avoid redefinition, we now include a precise definition in the revised version of this section.
>
> **”the accuracy drops from 1.00 to 0.02”**:
> Consider the C* context in Fig. 4a. If the green square (adjacent object) is treated as the ground truth for the query about the symbol “&” (first row), the average accuracy is 0.02. If the red circle (the grounded object) is treated as the ground truth, the accuracy is 0.98. This shows that the model relies on abstract identifiers rather than adjacent symbols.
> Notably, this **occurs without any fine-tuning**; it is an **emerging capability** of LVLMs. As discussed in Line 606, even the baseline input contains weak identifiers, and external cues reinforce this inherent capability (Line 612).
>
> &nbsp;
> ___
>
> ### Real-World Datasets:
>
>
> Causal mediation analysis is mainly conducted on simple and synthetic settings, even for LLMs. Conducting systematic observational and interventional analysis on real-world datasets is difficult because they are less controllable. In particular, activation-based causal mediation on natural images is hard to design, since transferring visual tokens across different contexts can introduce confounding factors due to the complexity of real-world datasets. However, we included an attention analysis on the MS-COCO dataset in Fig. 8. We extend this coarse-level observation by using ground-truth bounding box areas and computing attention and alignment values (as in Section 3) between these visual tokens and their corresponding generated words. The average attention and cosine similarity of true positive (TP) samples are reported in the following tablet:
> | Method| Mean Attention| Mean Cosine |
> |--|-|--|
> |Baseline| 0.24| 0.52|
> |Structured| 0.28| 0.54|
>
> Finally, we emphasize that for the MS-COCO dataset, the structure is not just a line but a grid with the number of partitions indicated in the center (Fig. 13). We provide various choices of external cues in Table 3.
>
> &nbsp;
> ___
>
> ### Point-by-Point Questions
>
> > 2. p. 3, line 122: “each token must be paired with …”
>
>  This refers to the analysis procedure. It is a post-processing step used to identify corresponding parts of the image and text (for example, the “red circle” token in the caption and its visual patch). After identifying these pairs, we compute attention or cosine similarity in a fine-grained manner.
>
> > 3. p. 3, line 124: “regular expression”
>
>  This refers to a standard regex method used to locate specific patterns in the text (for example, locating the phrase “red circle”).
>
> > 6. p. 5, line 223: “host context”
>
> This is the target context, which is a host for activations from the source context.
>
> > 7. p. 5, lines 217–218: swapping patch activations
>
> We patch the activations (residual streams) of the corresponding area from all layers. These activations are the final outputs of layers, so this step does not involve attention heads. Swapping is used because we need to transfer activations for two objects, not just one. For example, in Fig. 4a, transferring only the red circle from c' to c would give c* a red circle with the “&” grounding ID, while the green square from c would remain and also carry the “&” grounding ID. This would create ambiguity for analysis (two objects carry the “&” grounding ID). The swapping strategy avoids this issue.

---

### Official Review · Reviewer_QXed · 2025-10-30

**Soundness:** 2
**Presentation:** 3
**Contribution:** 2
**Rating:** 4
**Confidence:** 2

**Summary:**

This paper explores how simple visual cues can influence the spatial and grounding capabilities of large vision–language models (LVLMs). The authors introduce the concept of Grounding IDs—latent identifiers induced by external visual structures that help bind objects to their corresponding symbolic or textual components.

Through detailed representation and causal analyses, the study shows that these identifiers emerge as robust within-partition alignments in the embedding space and reduce the modality gap between image and text. Moreover, attention visualization and causal interventions reveal that Grounding IDs mediate cross-modal binding and improve grounding precision, effectively reducing hallucinations. Overall, the paper offers both theoretical insights into multimodal interpretability and practical improvements in multimodal robustness.

**Strengths:**

* The paper provides a novel conceptual contribution—the introduction of Grounding IDs—to explain how external cues can shape multimodal binding mechanisms.

* The authors perform detailed cross-attention and activation-swap experiments, demonstrating how symbolic annotations can directly modulate spatial perception and relational reasoning within LVLMs.

* The experimental methodology is carefully controlled, revealing causal links between visual structure and cross-modal alignment.

* The findings offer new interpretability insights, showing that external symbolic cues can serve as a controllable mechanism to enhance multimodal reasoning and reduce hallucinations.

**Weaknesses:**

1. The paper formatting deviates from the ICLR submission standard (notably excessive bottom margin spacing). I have encountered this issue in two of five papers I reviewed this year. Please fix it.

2. The motivation emphasizes precise grounding, yet the evaluation is limited to benchmarks such as CHAIR and POPE. Compared with works like VISER, this study lacks tasks involving object counting or visual grounding (e.g., referring expression comprehension, REC). Testing the proposed method on a stronger LVLM (e.g., Qwen-2.5VL) for these grounding benchmarks would make the claims more convincing.

**Questions:**

1. Are the Grounding IDs transferable across architectures or datasets, or are they model-specific phenomena?

2. Could the authors evaluate their method using a stronger LVLM such as Qwen-2.5VL to demonstrate generalizability?

3. How sensitive are the observed effects of Grounding IDs to the type and complexity of external cues (e.g., grid partitioning vs. text annotations)?

If these issues can be addressed or resolved, I would be willing to raise my overall score, as the paper presents an interesting and potentially impactful contribution to multimodal interpretability.

---

> ### Author Response · Authors · 2025-11-23
>
> We thank the reviewer for the detailed and thoughtful feedback. We are glad that the reviewer recognized the novelty of Grounding IDs, as well as their experimental design and interpretability insights.
>
> > **W1**:
>
> Thank you for pointing this out. The blank bottom margin caused us to lose usable space, which led to more compact explanations and brief descriptions that reduced the clarity. We have corrected it in the revised version, which also allowed us to expand several descriptions for clarity.
>
> &nbsp;
> ___
>
> > **W2**:
>
>
> We extend our experiments to two additional visual reasoning benchmarks (**counting** and **visual search**), which are directly related to visual grounding, and evaluate three different models: Qwen-3B, Qwen-7B, and GPT-4o.
>
>
> Counting Task:
>
>
> | Model | Grounding IDs | VISER | Base |
> |-----------|---------------|----------|---------|
> | Qwen-3B | 43.00% | 37.83% | 30.00% |
> | Qwen-7B | 53.00% | 43.33% | 29.67% |
> | GPT-4o | 32.33% | 26.50% | 10.50% |
>
>
> Visual Search:
>
>
>
> | Model | Grounding IDs | VISER | Base |
> |-----------|---------------|----------|---------|
> | Qwen-3B | 45.96% | 37.83% | 0% |
> | Qwen-7B | 52.25% | 40.00% | 30.00% |
> | GPT-4o | 80.62% | 73.40% | 49.41% |
>
>
> Our improvements come from using external cues with two key properties. First, each partition has a **unique** identifier (for example, a distinct character symbol for each row). Second, these cues appear in both the image and the prompt, which **aligns the two modalities**. Tables 1 and 3 provide ablations for these factors.
>
>
>
> However, we should clarify that despite the improved performance, the main goal of this study is to **understand** the mechanism by which aligned external structural cues improve grounding, rather than to introduce a method intended to compete with prior work. We focus on object hallucination mitigation as the primary behavioral evaluation since it’s a fundamental problem that directly involves grounding and has not been examined in earlier studies such as Rudman et al. (2025) and Izadi et al. (2025).
>
> &nbsp;
> ___
>
>
> > **Q1**:
>
>
> Transfer of Grounding IDs across different models is not fully straightforward, since architectures, training schemes, and tokenization may differ. However, they are general, and our behavioral analysis shows that both open-source and closed-source models benefit from their properties.
>
> Grounding IDs are also task-independent. In most LVLMs, visual tokens are processed before the prompt defines the task, meaning the visual embeddings and the induced identifiers are constructed prior to task conditioning.  As a result, Grounding IDs in the visual embeddings arise independently of the downstream task and can transfer across tasks. In our causal mediation analysis in Section 4, the source and target images could (in principle) come from different tasks (such as scene description, visual search, or counting). However, applying causal interventions across tasks requires careful experimental design to avoid unintended confounders.
>
> &nbsp;
> ___
>
> > **Q2**:
>
>
> For the behavioral analysis, we included both open-source models (LLaVA-1.5 and Qwen2.5-VL) and closed-source models (GPT-4o and Gemini-2.5-Pro) to assess consistency across different architectures (Lines 350, 355).
> Causal intervention analyses are also conducted on **Qwen2.5-VL**, which provides a stable open-source setting for these experiments.
>
> &nbsp;
> ___
>
>
> > **Q3**:
>
>
> We have included ablation studies in the main paper (Table 1, text-only and image-only conditions) and provided a more detailed analysis of cue types (numbers, letters, symbols, lines, grids) in Table 3 of the appendix. In the table below, we provided four new cues: *Symbol-only, Dots, Object Bounding Box (input-aware), and Symbols on Objects (input-aware)*. If you have additional suggestions, please let us know.
>
> | Variant                         | Precision | Recall | F1   | Accuracy |
> |----------------------------------|----------:|-------:|-----:|--------:|
> |Baseline               | 0.18      | 0.46   | 0.25   | 0.15    |
> | Numbers (no line)               | 0.53      | 0.43   | 0.47   | 0.31    |
> | Letters + Line                  | 0.64      | 0.48   | 0.55   | 0.38    |
> | Numbers + Line                  | 0.56      | 0.58   | 0.57   | 0.40    |
> | Symbols + Line                  | 0.67      | 0.50   | 0.58   | 0.47    |
> | Grid 4×4 Numbered               | 0.33      | 0.40   | 0.36   | 0.22    |
> | Grid 4×4 Numbered (no lines)    | 0.33      | 0.35   | 0.34   | 0.20    |
> | Numbers Right-Aligned + Line    | 0.39      | 0.55   | 0.46   | 0.30    |
> | Symbol-only                     | 0.48    | 0.50 | 0.49 | 0.33  |
> | Dots                     | 0.25    | 0.35 | 0.27 | 0.18  |
> | Object Bounding Box                      | 0.43    | 0.48 | 0.44 | 0.32  |
> | Symbols on Objects                 |   0.30       |  0.38     |   0.29    |  0.19      |

---

> > ### Comment · Reviewer_QXed · 2025-11-26
> >
> > Thanks for your response, most of my concerns have been addressed.
> >
> > However, the authors mentioned results about *two additional visual reasoning benchmarks (counting and visual search)*. While the results seem promising, the evaluation protocol and details are missing.
> >
> > Where do the benchmarks come from, and how are they defined? As I cannot find them in the main paper or appendix, I think these details shall be **specified**, and updated to the PDF file.
> >
> > I agree that the interpretability is important, but since prior works have demonstrated some findings on the visual cues, showing some practical value with performance income will make the paper stronger.

---

> ### Author Response · Authors · 2025-11-27
>
> Thank you for the follow-up comments.
>
> For the counting and visual search experiments, **we follow the VISER evaluation protocol**. The counting dataset contains synthetic 2D scenes with 10 to 20 circles, increasing in steps of 2, with 100 images per count level (600 total). The visual search dataset contains conjunctive search scenes with 20, 30, 40, and 50 objects. For each difficulty level, 100 images are generated: 50 with the target present and 50 with the target absent. Accuracy is computed for the visible and invisible cases separately, and we report the harmonic mean as in VISER.
>
> After your suggestion, **we revised Section 5**. It is now split into two subsections: hallucination mitigation and visual reasoning. The experimental setup and results for the reasoning benchmarks appear in the main paper (see Line 458 of the revised version). We can also further enrich this section by adding more models or additional reasoning benchmarks if helpful.
>
> We also added a new appendix section (A.9) that re-evaluates the results from Sections 3 and 4 using InternVL3.5 (8B) and Qwen2.5-VL (3B), which cover different architectures and scales compared to Qwen2.5-VL (7B).
> In addition, Appendix A.4 has been updated with new ablation baselines, including input-aware structural variants.
>
> We hope these revisions improve the completeness of the experimental evaluation.

---

### Official Review · Reviewer_ixV7 · 2025-11-03

**Soundness:** 3
**Presentation:** 2
**Contribution:** 2
**Rating:** 4
**Confidence:** 3

**Summary:**

The paper considers the relationship between external cues and multimodal binding in LVLMs. In particular, the paper shows that Grounding IDs (symbolic multimodal identifiers induced by external cues) can potentially strengthen attention between related components and improve multimodal grounding. Authors proposes that Grounding IDs are a key symbolic mechanism explaining how external cues enhance multimodal binding. Experiments and ablation studies are provided.

**Strengths:**

The strengths of the paper comes from the attempt to investigate how external cues (e.g., symbolic identifiers in this paper) can shape multimodal binding. The RQ of interest, how/why do external cues improve reasoning in LVLMs, is important and timely. The motivation from and connection to previous works (e.g., Binding IDs by Feng and Steinhardt, 2023, extension in LVLMs by Saravanan et al., 2025, VISER by Izadi et al., 2025) are presented relatively clearly. The central hypothesis (that structured symbols, either visually or textually, lead LVLMs to internally generate identifiers that bind objects to their designated partitions) is reasonable and aligns with takeaways of previous works.

**Weaknesses:**

### Writing

The paper can benefit from further improvements on the writing itself and clarifying on the interpretation of presented results (detailed in "Questions" section).

### Novelty and Significance

There is a concern on the novelty and significance of the proposed Ground IDs (detailed in "Questions" section).

**Questions:**

Questions and comments:

1. The paper can get difficult to parse from time to time. For instance, the interpretation of the figures are not easy to follow. I am not entirely convinced about authors' interpretation (of Fig. 5) that the model shifts toward predicting the intervened grounded object in the later layers (20--27), now that the similar uptick trends are observed both in baseline and structured (in Fig. 3a). The Fig. 6 can also benefit from additional clarifications (e.g., how each entry is calculated precisely), beyond lines 272--279.

1. The sentence "The model is tasked with generating ..." was repeated twice (lines 119--120).

1. The concern on novelty and significance. Overall, the results confirm previous claims (e.g., Schrodi et al., 2024, line 138), by extending scaffolds from horizontal lines to symbols. While I understand that the (color-coded) symbols may not be explicitly addressed by previous works (e.g., Saravanan et al., 2025), the effectiveness of textual references/cues are well-established. The RQ of interest itself (line 94, why do external cues improve reasoning in LVLMs) is important and timely, but the claimed benefits of Grounding IDs do not seem to be significant advancement over existing literature.

---

> ### Author Response · Authors · 2025-11-23
>
> We thank the reviewer for the thoughtful feedback and for highlighting the relevance of our research question and the clarity of our motivation and hypothesis. We address the raised concerns below:
>
>
>
> > **Q1**
>
> We agree that in some parts the interpretation and clarifications were too brief. This was mainly due to the strict space limit of the 9-page submission which required us to shorten explanations and figure captions. In the revised version, we have expanded these sections and added full descriptions where needed.
>
>
> ### Fig. 5 and its relation to Fig. 3:
>
>
> The analysis computes the layer-wise logit difference between the grounded object and the adjacent object. The setting matches the patched-context experiment in Fig. 4a, and the goal is to identify from which layer the model shifts its prediction toward the grounded object. This provides a complementary causal observation to the representational result in Fig. 3a, where alignment for structured examples diverges from the baseline in the later layers.  Indeed, these findings show that the increase in image-text alignment for structured samples in the late layers (Fig. 3) is accompanied by causal evidence that the model relies on internal grounded IDs when producing the final output (Fig. 5).
>
>
> We also clarify that the conclusion of Fig. 5 should not be interpreted through a comparison with the **baseline model**, because the experiment is completely defined **under the structured setting**  (see Fig. 4). The appropriate comparison is between grounded and adjacent objects in the patched context (c*), which is why we use the logit difference.
>
>
> We think the reviewer’s question about the baseline may relate to a possible confounding factor that the model generally tends to refine its prediction in later layers. However, our analysis focuses on the difference between two valid responses rather than their absolute logits, which completely eliminates the effect of this confounder.
>
>
> ### Regarding Fig. 6:
>
> We agree that the explanation was too compact, partly because the figure had to be placed inline to save space. In the revised version, we make this part clearer by expanding the description of how the differential vectors and cosine similarities are computed.
>
> &nbsp;
> ___
>
> > **Q2**
>
> We thank the reviewer for pointing this out. The repeated sentence has been removed in the revised version.
>
> &nbsp;
> ___
>
> > **Q3**
>
> We clarify that the idea of our paper is clearly far from Schrodi et al. (2024). We cite it in line 138 not for its contribution to reducing the modality gap (they do not claim this), but because they conduct a comprehensive study of the modality gap in small VLM architectures such as CLIP. We make the citation purpose clearer in the revised version. Indeed, Schrodi et al. (2024) also report that post-hoc removal of gap-contributing dimensions distorts the embedding geometry and hurts performance. In contrast, we show that external multimodal cues reduce the modality gap through the internal dynamics of LVLMs and, as a consequence, enhance visual grounding and downstream task performance. **Showing this kind of reduction in the modality gap without harmful interventions (such as removing dimensions) is itself a novel finding of our work**.
>
>
> The novelty of our paper also differs from prior interpretability works that study abstract components and mechanisms in LLMs and LVLMs. We acknowledge the relation between Grounding ID and the notion of binding (Feng & Steinhardt, 2023; Saravanan et al., 2025), since grounding is a special type of binding, and we state this repeatedly (Lines 16, 69, 97, etc) including in the paper title. The mentioned works focus on the study of the binding between items in standard inputs (*intrinsic*). However, **we focus on the *extrinsic* effect of paired external cues that are provided in both modalities**; these are not just color-coded symbols, as our ablations (Table 1) show that unimodal cues are not effective enough. More importantly, we investigate how these multimodal external cues themselves participate in multimodal binding, how this imposed binding changes the internal dynamics of the model, and finally, how this enhances visual reasoning.
>
>
> We refer to this as Grounding IDs because we observe **they strengthen attention and alignment and reinforce the model’s inherent grounding ability**. Furthermore, our primary question is to **understand** why system-2-style approaches that use such external cues lead to performance gains (also noted by the reviewer). We use this special type of binding as a tool to interpret this behavior.
>
> Finally, while our method is mainly focused on understanding, it also introduces improved structural cues that enhance downstream performance. Aligned multimodal cues were not investigated in previous studies such as VISER. Furthermore, the object hallucination mitigation part itself demonstrates a standalone methodological contribution that has not been examined by prior work.

---

### Author Response · Authors · 2025-12-03
**Clarifications on Reviewers' Comments**

Dear Meta-Reviewers and Area Chairs, thank you for your time and consideration.

As you saw, only one brief comment (from reviewer QXed) appeared during the discussion phase, and we had no opportunity to receive feedback from the other reviewers.
 We believe several misunderstandings arose due to a cursory reading of the paper. For example, “the novelty concern” raised by ixV7, “the request for models that were already evaluated” by QXed, “questions about basic experimental concepts such as the baseline or activation patching” raised by Q3Bc, and “the request for ablation studies already included in the paper” noted by Gyen, are a few examples of overlooking the original script by reviewers. We clarified all of these points in our responses with proper references to the paper.

Our behavioral experiments were already conducted on multiple open-source and closed-source models, demonstrating the generality of Grounding IDs. Notably, we show that Grounding IDs can also enhance grounding capability in large proprietary models such as GPT-4o and Gemini 2.5-Pro, which are not commonly evaluated by hallucination-mitigation studies. During the rebuttal, we further demonstrated the effectiveness of Grounding IDs by expanding experiments with new models, new benchmarks, and new ablations. While the main contribution of our paper lies in the *understanding* of GroundingID, we conducted additional behavioral experiments requested by Reviewers QXed and Gyen, and all behavioral experiments demonstrate consistent improvements across diverse settings.

In addition to the novelty clarification provided in response to reviewer ixV7, we believe that our approach offers useful insight for interpretability research in multimodal models. While the LLM literature includes many interpretability studies, multimodal models still lack interpretability work with causal analysis. We think that our research methodology — including the research question and hypothesis, the experimental design (both observational and behavioral), and the causal mediation analysis — can benefit future interpretability studies in this area.

---

### Meta-Review · Area_Chair_KTNP · 2025-12-28

**Summary:**

This submission received negative scores (4444). The majors concerns raised by reviewers are about novelty of the approach and strength of empirical validation. After discussion, responses to major concerns from one of the reviewer were not responded. One of the reviewers responded to the rebuttal during the discussion phase but raised new concerns about the evaluation protocol and details. After carefully reading the paper, the review comments and the discussions, I suggest the authors to carefully consider the concerns raised by reviewers and submit their work to the next conference.

**Reviewer Concerns:**

The authors provided responses to the major concerns raised by the reviewers. One of the reviewers responded to the rebuttal during the discussion phase but raised new concerns about the evaluation protocol and details.

**Reviewer Scores:**

One of the reviewers responded to the rebuttal during the discussion phase, but there is no evidence to show all reviewers assigned negative scores would have changed their scores.

---

### Decision · Program_Chairs · 2026-01-26

Reject